

# Transfer of diazotroph-derived nitrogen to the planktonic food web across gradients of N$_2$ fixation activity and diversity in the Western Tropical South Pacific

Mathieu Caffin[1], Hugo Berthelot[1,2], Véronique Cornet-Barthaux[1], Sophie Bonnet[1,3]

[1]Aix Marseille Université, CNRS, Université de Toulon, IRD, OSU Pythéas, Mediterranean Institute of Oceanography (MIO), UM 110, 13288, Marseille, France
[2] Laboratoire des sciences de l'environnement marin, IUEM, Université de Brest-UMR 6539 CNRS/UBO/IRD/ Ifremer, Plouzané, France
[3] Aix Marseille Université, CNRS, Université de Toulon, IRD, OSU Pythéas, Mediterranean Institute of Oceanography
(MIO), UM 110, 98848, Nouméa, New Caledonia

*Correspondence to*: Mathieu Caffin (mathieu.caffin@mio.osupytheas.fr)

**Abstract.**

Biological dinitrogen (N$_2$) fixation provides the major source of new nitrogen (N) to the open ocean, contributing more than
atmospheric and riverine inputs to the N supply. Yet the fate of the diazotroph-derived N (DDN) in the planktonic food web is poorly understood due to technical limitations. The main goals of this study were to i) quantify how much of DDN is released to the dissolved pool during N$_2$ fixation and how much is transferred to bacteria, phytoplankton and zooplankton, ii) to compare the DDN release and transfer efficiencies under contrasting N$_2$ fixation activity and diversity the oligotrophic waters of the Western Tropical South Pacific (WTSP) Ocean. We used nanometer scale secondary ion mass spectrometry
(nanoSIMS) coupled with $^{15}$N$_2$ isotopic labelling and flow cytometry cell sorting to track the DDN transfer to plankton, in regions were the diazotroph community was either dominated by *Trichodesmium* or by UCYN-B. After 48 h, ~20-40 % of the N$_2$ fixed during the experiment was released to the dissolved pool when *Trichodesmium* dominated, while the DDN release was not quantifiable when UCYN-B dominated. ~7-15 % of the total fixed N (net N$_2$ fixation + release) was transferred to non-diazotrophic plankton within 48 h, with higher transfer efficiencies (15 ± 3 %) when UCYN-B dominated
as compared to when *Trichodesmium* dominated (9 ± 3 %). Most of the DDN (>90 %) was transferred to picoplankton (*Synechococcus*, *Prochlorococcus* and bacteria) in all experiments. The cyanobacteria *Synechococcus* and *Prochlorococcus* were the primary beneficiaries (~65-70 % of the DDN transfer), followed by heterotrophic bacteria (~23-34 % of the DDN transfer). The DDN transfer in bacteria was the highest (34 ± 7 %) when UCYN-B were dominating the diazotroph community. Regarding higher trophic level, the DDN transfer to the dominant zooplankton species was more efficient when
the diazotroph community was dominated by *Trichodesmium* (~5-9 % of the DDN transfer) than when it is dominated by UCYN-B (~28 ± 13 % of the DDN transfer). To our knowledge, this study provides the first quantification of DDN release and transfer to phytoplankton, bacteria and zooplankton communities in open ocean waters. It reveals that despite UCYN-B



fix $N_2$ at lower rates compared to *Trichodesmium* in the WTSP, the DDN from UCYN-B is much more available and efficiently transferred to the planktonic food web than the DDN coming from *Trichodesmium*.

## 1 Introduction

Nitrogen (N) is one of the basic building blocks of life, though much of the global Ocean surface (~70 %) is oligotrophic and
characterized by low N availability, which limits primary productivity and phytoplankton growth (Falkowski, 1997; Moore et al., 2013). In these N-depleted areas of the tropical and subtropical ocean, biological dinitrogen ($N_2$) fixation (the reduction of atmospheric $N_2$ into bioavailable ammonia) sustains the major part of new production and organic matter export (Bonnet et al., 2009; Caffin et al., 2017; Capone et al., 2005; Karl et al., 2012). At the global scale, $N_2$ fixation is the major source of new N to the ocean, before atmospheric and riverine inputs (100-150 Tg N $yr^{-1}$, Gruber, 2008). $N_2$ fixation is
performed by prokaryotic organisms termed diazotrophs, which include the non-heterocystous filamentous cyanobacterium *Trichodesmium* (Capone et al., 1997; Carpenter, 1983), heterocystous cyanobacteria living in symbiosis with diatoms (or diatom-diazotroph associations, termed DDAs; Villareal, 1994), unicellular cyanobacteria termed UCYN (subdivided in Group A, B and C based on the *nifH* gene sequence, Zehr et al., 1998, 2008; Zehr and Turner, 2001), and diverse non-cyanobacterial bacteria (Bombar et al., 2015; Moisander et al., 2014; Riemann et al., 2010). Although considerable efforts
have been deployed over the past decades to quantify $N_2$ fixation, identify the major players, and assess their biogeographical distribution in relation with environmental drivers, the fate of new N provided by $N_2$ fixation in the ocean, its role on the planktonic food web structure and large-scale biogeochemical fluxes is still poorly understood.

Early studies have reported high dissolved organic N (DON) and ammonia ($NH_4^+$) concentrations during and following *Trichodesmium* blooms in the Indian Ocean (Devassy et al., 1978, 1979; Glibert and O'Neil, 1999), suggesting
that *Trichodesmium* release part of the recently fixed $N_2$ (hereafter referred to as diazotroph-derived N, DDN) to the dissolved pool, which could subsequently be consumed by the surrounding plankton communities. The first direct release measurements were performed in the early 1990s and showed that *Trichodesmium* colonies isolated from the tropical Atlantic Ocean release ~50 % of the recently fixed $N_2$ (Glibert and Bronk, 1994). Accumulations of DON and $NH_4^+$ have subsequently been confirmed near *Trichodesmium* blooms in the Pacific (Karl et al., 1992, 1997) and Atlantic Oceans (Lenes
et al., 2001), although not systematic (Bonnet et al., 2016a; Hansell and Carlson, 2001) , and in senescent *Trichodesmium* cultures (Mulholland and Capone, 2000), possibly related to the *Trichodesmium* programmed cell death (PCD, Berman-Frank et al., 2004). This DDN release has been attributed i) to endogenous processes such as the dissipation of excess electrons linked to an excess of light (Wannicke et al., 2009) or to a means for the filamentous diazotrophs to transfer fixed N from N2-fixing cells to vegetative cells (Mulholland and Capone, 2000), ii) to exogenous processes such as viral lysis
(Hewson et al., 2004; Ohki, 1999) or 'sloppy feeding' by copepods (O'Neil, 1999).

Numerous studies performed in culture (Hutchins et al., 2007; Karl et al., 1992, 1997) and in the field (Benavides et al., 2013b; Konno et al., 2010; Mulholland and Bernhardt, 2005) focused on quantifying this release, and most of them were



performed on *Trichodesmium* and were based on the difference between the measurement of gross $N_2$ fixation (through the acetylene reduction method, Capone (1993)) and net $N_2$ fixation rates (through the $^{15}N_2$ isotope labelling method, Montoya et al. (1996)) (Mulholland et al., 2004). It was thus shown that the DDN released to the dissolved pool averages ~50 % (10 to 80 %) of total $N_2$ fixation. The estimates based on this approach have then be questioned since the discovery of the

methodological underestimation of net $^{15}N_2$ rates when the $^{15}N_2$ tracer is injected as a bubble in the incubation bottles (Mohr et al., 2010), leading to a potential overestimation of the DDN release. The application of the new method in which the $^{15}N_2$ is added as dissolved in a subset of seawater previously $N_2$ degassed (Mohr et al., 2010) has recently shown in *Trichodesmium* cultures that the DDN release represents less than 1 % of total $N_2$ fixation (Berthelot et al., 2015). An alternative approach based on the direct measurement of the $^{15}N$ enrichment of both the particulate and dissolved pools

(Glibert and Bronk, 1994; Slawyk and Raimbault, 1995) after incubation with $^{15}N_2$ present the advantage of providing ratio of particulate $N_2$ fixation versus DDN release, without being affected by potential underestimations issues. The studies based on this approach reveal that the proportion of DDN released to the dissolved pool ranges from 10 % to >80 % of total $N_2$ fixation measured in field (Benavides et al., 2013b; Berthelot et al., 2017; Glibert and Bronk, 1994; Konno et al., 2010). The release appears to be higher when *Trichodesmium* dominate the diazotroph community (Berthelot et al., 2017; Bonnet et al.,

2016a; Glibert and Bronk, 1994) than when UCYN dominate (Benavides et al., 2013a; Bonnet et al., 2016b). The DDN released to the dissolved pool measured by this direct approach is generally much lower in culture studies (<5 %) (Benavides et al., 2013a; Berthelot et al., 2016), suggesting that external factors such as sloppy feeding and viral lysis have a strong influence on the DDN release by diazotrophs in field.

The DDN released in the surface ocean is potentially available for surrounding planktonic communities, but its fate

in the planktonic food web has poorly been quantified mainly due to methodological locks. Devassy et al. (1979) who reported high DON and $NH_4^+$ concentrations near *Trichodesmium* blooms were also the first to observe that during the decline of the *Trichodesmium* blooms, diatom abundances increased, followed by a succession of cladocerans, dinoflagellates, green algae and finally copepods. In the tropical Atlantic Ocean, high abundances of non-diazotrophic phytoplankton have also been observed following *Trichodesmium* blooms (Mulholland et al., 2004), and more recently,

Chen et al. (2011) showed a positive correlation between abundances of *Trichodesmium* and diatoms in the Kuroshio Current. These studies suggest a link between diazotroph blooms and non-diazotrophic organisms. Studies based on size-fractionation carried out during *Trichodesmium* bloom incubated in the presence of $^{15}N_2$ report that ~10 % of the fixed $N_2$ by *Trichodesmium* (recovered in the size fraction >30 μm) was rapidly transferred to non-diazotrophic organisms (recovered in the <30 μm fraction; Bryceson and Fay (1981)). Using similar methods, other studies suggested that during *Trichodesmium*

blooms (Garcia et al., 2007; Mulholland et al., 2004) and *Nodularia* and *Aphanizomenon* blooms (Ohlendieck et al., 2000), 5 to 10 % of the DDN is transferred to the picoplankton compartment. However, the methods based on size fractionation do not discriminate the DDN transfer towards the picoplankton to the $N_2$ fixation carried out by this same picoplankton (in particular the UCYN-A, one of the most abundant diazotroph in our ocean (Luo et al., 2012)). This method therefore potentially overestimates the DDN transfer and is not applicable to study the DDN transfer associated with UCYN.



Moreover, it is not possible with size fractionation methods to determine which populations (e.g. autotrophic vs. heterotrophic plankton, small vs. large plankton) have benefited the most from this source of new N. The recent use of high-resolution nanometer scale secondary ion mass spectrometry (nanoSIMS) coupled with $^{15}N_2$ isotopic labeling and flow cytometry cell sorting (Berthelot et al., 2016; Bonnet et al., 2016a, 2016b) has proved its efficiency in the quantification of the DDN transfer to specific groups of phytoplankton and bacteria in the coastal Western Tropical South Pacific (WTSP) and role of $N_2$ fixation by the colony-forming cyanobacterium *Aphanizomenon* in the Baltic Sea (Adam et al., 2016). Using this method in *Trichodesmium* blooms in the coastal WTSP, Bonnet et al. (2016a) revealed that after 48 h $13 \pm 2$ % to $48 \pm 5$% of the fixed $N_2$ was released to the dissolved pool and $6 \pm 1$ % to $8 \pm 2$ % of this DDN was transferred to non-diazotrophic plankton, mainly diatoms ($45 \pm 4$ % to $61 \pm 38$ %) and bacteria ($22 \pm 27$ % to $38 \pm 12$ %). A mesocosm experiment performed in the New Caledonian lagoon during a UCYN-C bloom (Bonnet et al., 2016b) revealed after 48 h $16 \pm 6$ % of the fixed $N_2$ was released to the dissolved pool and $21 \pm 4$ % of this DDN was transferred to non-diazotrophic plankton, mainly picoplankton ($18 \pm 4$ %) and diatoms ($3 \pm 2$ %). Finally, a comparative study between *Trichodesmium* vs. UCYN, performed from culture isolates (Berthelot et al., 2016), revealed that the DDN transfer to non-diazotrophic plankton is twice as high with *Trichodesmium* as with UCYN. To date, the transfer of DDN to different groups of plankton from different diazotroph (*Trichodesmium* vs. UCYN) in the open ocean has never been investigated.

Regarding higher trophic levels, the low δ15N signature of zooplankton reveals that $N_2$ fixation can significantly fuel zooplankton growth in high $N_2$ fixation areas (Aberle et al., 2010; Landrum et al., 2011; Loick-Wilde et al., 2012; Mompeán et al., 2013; Montoya et al., 2002; Sommer et al., 2006; Wannicke et al., 2013). Few studies reports active grazing of *Trichodesmium* by some specific copepods (Micro- and Macrosetella, O'Neil et al. (1996); O'Neil and Roman (1992)). However, *Trichodesmium* has been shown to be toxic for most of the grazers (Guo and Tester, 1994; Hawser et al., 1992; Hawser and Codd, 1992) and the low δ15N signature found in zooplankton (indicative of DDN consumption) where *Trichodesmium* thrive most likely originates from indirect transfer mediated by recycling processes (Capone et al., 1994; Capone and Montoya, 2001; Letelier and Karl, 1996) rather than direct grazing. A recent study based on $^{15}N_2$ labelling in the coastal WTSP (Hunt et al., 2016) reveals that the DDN is less efficiently transferred to zooplankton when *Trichodesmium* and DDA dominate the diazotroph community than when UCYN-C dominate, suggesting that the DDN transfer efficiency to zooplankton strongly depends on the diazotroph involved in the $N_2$ fixation. To our knowledge, this has never been investigated in the open ocean.

The WTSP Ocean has recently been identified as a hot spot of $N_2$ fixation (Bonnet et al., 2017) and is characterized by trophic and $N_2$ fixation gradients (Moutin et al., 2017), with oligotrophic waters characterized by high $N_2$ fixation rates ($631 \pm 286$ µmol N m$^{-2}$ d$^{-1}$) mainly associated with *Trichodesmium* in the western part, and ultra-oligotrophic waters characterized by low $N_2$ fixation rates ($85 \pm 79$ µmol N m$^{-2}$ d$^{-1}$) mainly associated with UCYN in the eastern part (Bonnet et al., this issue; Stenegren et al., this issue). We performed a series of experiments under contrasting situations (either when *Trichodesmium* or UCYN was dominating the diazotroph community) to study the fate of DDN in the planktonic food web, with the following specific goals: (1) quantify the proportion of DDN released to the dissolved pool relative to total $N_2$



fixation, (2) quantify the DDN transfer to the non-diazotrophic phytoplankton and bacteria, and (3) quantify the DDN transfer to zooplankton.

## 2 Material and Methods

### 2.1 Experimental setup for DDN transfer experiments in phytoplankton and heterotrophic bacteria

This study was carried out during the OUTPACE (Oligotrophic to UlTra oligotrophic PACific Experiment) cruise (DOI: http://dx.doi.org/10.17600/15000900) onboard the R/V *L'Atalante* in February-March 2015 (austral summer). Samples were collected along a ~4000 km west to east zonal transect along ~19°S starting in New Caledonia and ending in French Polynesia, crossing Melanesian archipelago waters (hereafter referred to as MA waters) around New Caledonia, Vanuatu, Fiji up to Tonga and South Pacific Gyre waters located at the western boundary of the South Pacific Gyre (hereafter referred

to as GY waters) (see Moutin et al. (2017) for details). Three experiments are reported here (hereafter named E1, E2 and E3). Two were performed at stations located in MA waters: station LD A : 19°12.8'S - 164°41.3'E, and station LD B: 18°14.4'S - 170°51.5'W, where *Trichodesmium* accounted for 95 and 100 % of the total diazotroph community quantified by quantitative PCR, Stenegren et al. (2017)). The third experiment was performed in GY waters: station LD C: 18°25.2'S - 165°56.4'W, where UCYN accounted for 82 % of the total diazotroph community, Stenegren et al. (2017)).

15       The experiments were designed according to Berthelot et al. (2016) and Bonnet et al. (2016a). For experiments E1 and E2, seawater was collected by the underway pumping system at 6 m-depth. For E3, seawater was collected at 55 m-depth using Niskin bottles mounted on a CTD rosette. For all experiments, seawater was collected into 8 HCl-washed-sample rinsed (three times) 4.5 L polycarbonate bottles equipped with septum caps. 5 mL of $^{15}N_2$ gas (98.9 atom% $^{15}N$, Cambridge isotopes) were injected into all bottles using a gas-tight syringe. The purity of the $^{15}N_2$ Cambridge isotopes stocks

was previously checked by Dabundo et al. (2014) and more recently by Benavides et al. (2015) and Bonnet et al. (2016b), who concluded that the purity is satisfying (2 x $10^{-8}$ mol:mol N of $^{15}N_2$) and therefore do not alter the results presented below. The bottles were shaken 30 times to facilitate the $^{15}N_2$ dissolution and, except for the T0 set of bottles (see below), were incubated for 48 h in on-deck incubators covered with blue screening (50 % surface irradiance for E1 and E2 and 15 % surface irradiance for E3) and cooled with circulating surface seawater. At T0 and after incubation, a set of 4 bottles were

collected and subsampled for the following measurements (see below for methods): $N_2$ fixation rates, DDN release, quantification of diazotrophs, heterotrophic bacteria and pico-, nano- and microphytoplon enumeration, organic and inorganic nutrient concentrations, and $^{15}N$-enrichment on diazotrophs and non diazotrophic plankton. Unless otherwise stated, each parameter reported below was measured in triplicates.

### 2.2 Net $N_2$ fixation rates

30       $N_2$ fixation rates were measured using the $^{15}N_2$ isotopic tracer technique (adapted from Montoya et al. (1996)). The $^{15}N_2$ bubble technique was intentionally chosen to avoid any potential overestimation due trace metal and dissolved organic





matter (DOM) contaminations often associated with the preparation of the $^{15}N_2$-enriched seawater (Klawonn et al., 2015; Wilson et al., 2015) in our incubation bottles as Fe and DOM have been seen to control $N_2$ fixation or *nifH* gene expression in this region (Benavides et al., 2017; Moisander et al., 2012). However, the $^{15}N$ enrichment of the $N_2$ pool available for $N_2$ fixation was measured in all incubation bottles to ensure accurate rate calculations. Briefly, 12 mL were subsampled after

incubation into Exetainers® fixed with $HgCl_2$ (final concentration 20 mg $L^{-1}$) that were preserved upside down in the dark at 4°C until analyzed using a membrane inlet mass spectrometer (MIMS) according to Kana et al. (1994).

At the end of incubations, 2.3 L of the triplicates 4.5 L amended bottles were gently filtered onto pre-combusted (450 °C, 4 h) Whatman GF/F filters (25 mm diameter, 0.7 μm nominal porosity). Filters were stored in pre-combusted glass vials at -20 °C during the cruise, and then dried at 60 °C for 24 h before analysis onshore. $^{15}N$-enrichments of particulate N

collected on filters were determined using an Elemental Analyzer coupled to an Isotope Ratio Mass Spectrometer (EA-IRMS, Integra2 Sercon Ltd). The accuracy of the EA-IRMS system was systematically controlled using International Atomic Energy Agency (IAEA) reference materials, AIEA-N-1 and IAEA-310A. In addition, the $^{15}N$-enrichement of the ambient (unlabeled) particulate N was measured at each station at T0 and was used as the "initial" $^{15}N$-enrichment as termed in Montoya et al. (1996).

**2.3 DDN released to the dissolved pool**

The DDN released to the dissolved pool under the form of $NH_4^+$ and DON during the $N_2$ fixation process was quantified using the three step diffusion method extensively described in Berthelot et al. (2015) and derived from Slawyk and Raimbault (1995). This method enables the differentiation of the nitrate ($NO_3^-$), $NH_4^+$ and DON pools, and measured their respective $^{15}N$-enrichment. As the $NO_3^-$ pool was negligible, the total dissolved N (TDN) pool was defined as the sum of

DON and $NH_4^+$ pools (TDN = DON + $NH_4^+$). After incubation with the $^{15}N_2$ tracer, 300 mL of the filtrate passed through pre-combusted Whatman GF/F filters were collected in 500 mL Duran Schott borosilicate flasks, poisoned with $HgCl_2$ (300 μL, final concentration 20 mg $L^{-1}$) and stored at 4°C in the dark until analysis. At the end of each step, $NH_4^+$ and DON fraction were recovered on acidified pre-combusted Whatman GF/F filters, dried 24 h at 60 °C and stored in pre-combusted glass tubes until analysis on EA-IRMS as described in Berthelot et al., (2015). The DDN release was the sum of all forms of

N released in each fraction.

**2.4 Inorganic and organic nutrient analyses**

$NH_4^+$ concentrations were measured fluorimetrically according to Holmes et al. (1999) on a FP-2020 fluorimeter (Jasco, detection limit = 3 nM). $NO_3^-$ and nitrite ($NO_2^-$) concentrations were measured using standard colorimetric procedures (Aminot and Kérouel, 2007) on a AA3 AutoAnalyzer (Seal-Analytical). DON concentrations were measured by the wet

oxidation method according to Pujo-Pay and Raimbault (1994). After the wet oxidation, the concentration of TDN was measured on a AA3 AutoAnalyzer (Seal-Analytical). DON concentrations were obtained by difference between TDN and Dissolved Inorganic N ($NH_4^+$ + $NO_3^-$ + $NO_2^-$) measured in parallel.





## 2.5 Plankton abundance determination

The abundance of *Trichodesmium* and UCYN-B was determined microscopically: 2.2 L of each triplicates $^{15}N_2$-amended 4.5 L bottles were gently filtered onto 2 µm pore size, 25 mm diameter Millipore polycarbonate filter, fixed with paraformaldehyde (2 % final concentration) for 1 h. *Trichodesmium* were enumerated on the entire surface of the filter at x100 magnification with a Zeiss Axio Observer epifluorescence microscope fitted with a green (510-560 nm) excitation filter. The number of cells per trichomes was counted on 20 trichomes for each experiment; we counted an average of 85 and 115 cells trichome$^{-1}$ for E1 and E2, respectively. UCYN-B were counted on 40 fields (1.3 mm$^2$ fields; 0-2800 UCYN-B per field) scanned and analyzed with the ImageJ1 software.

Samples for micro-phytoplankton identification and enumeration were collected in each of the triplicate 4.5 L incubated bottles (except for E1 were only one replicate was available) in five 50 mL sterile PP tubes and preserved in acidic Lugol's solution (0.5 % final concentration). Diatoms were enumerated from a 250 mL subsample following the Utermohl method (Hasle, 1978), using a Nikon TE2000 inverted microscope equipped with phase-contrast and a long distance condenser. Diatoms were identified to the lowest possible taxonomic level in one of the three replicates.

Pico, nano-phytoplankton and heterotrophic bacteria abundances were determined by flow cytometry. After incubation, 1.8 mL were subsample from triplicate 4.5 L bottles into cryotubes, fixed with paraformaldehyde (200 µL, 4 % final concentration), flash-frozen in liquid $N_2$, and stored at -80 °C until analysis on a FACSCalibur (BD Biosciences, San Jose, CA) according to Marie et al. (1999), at the PRECYM flow cytometry platform (https://precym.mio.univ-amu.fr/). Before analysis, samples were thawed at room temperature in the dark. For bacterial abundance, 300 µL of each sample was incubated with SYBR Green II (Molecular Probes, 0.5 % final concentration) for 15 min in the dark. For phytoplankton and bacterial abundances measurements, before analysis, 2 µm beads (Fluoresbryte, Polysciences) and Trucount beads (BD Biosciences) were added to the samples. Analyses were run during 1.5 and 3 min at high and medium flow for phytoplankton and bacteria, respectively. The 2 µm beads were used as an internal control and to discriminate picophytoplankton (< 2 µm) from nanophytoplankton (> 2 µm) populations. Phytoplankton communities were clustered as *Prochlorococcus* spp. cell like, *Synechococcus* spp. cell like, nano-eukaryotes cell like, pico-eukaryotes cell like, and UCYN-B cell like. The Trucount beads were used to determine the volume analyzed. All data were acquired using the CellQuest software (BD Biosciences), and data analysis was performed using the SUMMIT v4.3 software (Dako).

## 2.6 Cell sorting and sampling for nanoSIMS analyses

For flow cytometry cell sorting and subsequent analysis using nanoSIMS, 1 L of one of the 4.5 L bottles was filtered onto 0.2 µm pore size 47 mm polycarbonate filters to preconcentrate the cells and facilitate cell sorting. Filters were placed in a 5 mL cryotube® filled with 0.2 µm filtered seawater with PFA (2% final concentration), and incubated 1 h at room temperature in the dark. The cryovials were vortexed, ~20 s to detach the cells from the filter and were stored at -80 °C until analysis. Cell sorting was performed using a Becton Dickinson Influx™ (BD Biosciences, Franklin Lakes, NJ) high speed



cell sorter of PRECYM platform, as described in Bonnet et al. (2016b). Planktonic groups (Pico, nano-phytoplankton, bacteria and UCYN-B (for E3 only)) were separated using the same clusters as for the phytoplankton abundance determination. At the issue of the cell sorter, the cells were directly dropped on a 0.2 μm pore size, 13 mm diameter polycarbonate filter (Millipore) connected to a low pressure pump in order to concentrate them on a surface as small as
possible. The filters were stored at -80 °C until nanoSIMS analyses.

To recover large phytoplanktonic cells (*Trichodesmium* and large diatoms), 1 L of the same 4.5 L bottle was filtered on 10 μm pore size 25 mm polycarbonate filters. The cells were fixed with PFA (2 % final concentration), incubated 1 h at ambient temperature and stored at -20 °C until nanoSIMS analyses.

## 2.7 NanoSIMS analyses

Just before nanoSIMS analyses, filters were thawed at ambient temperature and sputtered with gold and palladium to ensure conductivity. Analyses were performed on a NanoSIMS N50 (Cameca, Gennevilliers, France) at the French National Ion MicroProbe Facility as previsouly described in Bonnet et al. (2016a, 2016b) and Berthelot et al. (2016). Briefly, high density cells area were retrieved using the nanoSIMS optical camera (Fig.1 f.). Samples were pre-sputtered with prior to analyses for at least 2 min to remove surface contaminants and increase conductivity with a ~22 pA Cesium primary beam. For the
analysis, a ~1.2 pA Cesium (16 KeV) primary beam focused onto ~100 nm spot diameter was scanned across a 256×256 or 512×512 pixel raster (depending on the image size, which ranged from 20 μm × 20 μm to 40 μm × 40 μm) with a counting time of 1 ms per pixel. Negative secondary ions ($12C-$, $13C-$, $12C14N-$, $12C15N-$ and $28Si-$) were collected by electron multiplier detectors, and secondary electrons were imaged simultaneously. A total of 20 serial quantitative secondary ion images were generated to create the final image. Mass resolving power was ~8000 in order to resolve isobaric interferences.
Data were processed using the LIMAGE software. Briefly, all scans were corrected for any drift of the beam and sample stage during acquisition. Isotope ratio images were created by adding the secondary ion counts for each recorded secondary ion for each pixel over all recorded planes and dividing the total counts by the total counts of a selected reference mass. Individual cells were easily identified in nanoSIMS secondary electron, 12C-, 12C14N- and 28Si images that were used to define regions of interest (ROIs) around individual cells. A total of ~1500 ROI was analyzed. For each ROI, the [15]N-
enrichment was calculated. 50 to 200 cells were analyzed for each plankton group and for each experiment.

## 2.8 Cell-specific N content and DDN transfer calculations

To determine cell-specific N contents, cell sizes of *Trichodesmium* and dominant diatoms were directly measured on each sample collected for microscopy (see section 2.5). For *Trichodesmium*, cell length and width were measured on 25 to 50 cells per sample at x400 magnification with a Zeiss Axio Observer epifluorescence microscope. For diatoms, the cell cross
section and apical and transapical dimensions were measured on at least 20 cells using a Nikon TE2000 inverted microscope equipped with phase contrast and a long-distance condenser. For UCYN-B, cells diameters were directly measured on the nanoSIMS images. The biovolumes (BV) of *Trichodesmium*, UCYN-B and diatoms were estimated following the geometric





model of each cell type (Sun and Liu, 2003). The cellular carbon (C) contents were determined by using the relation between BV and C content according to Verity et al. (1992) for *Trichodesmium* and UCYN-B, and according to Eppley et al. (1970) and Smayda (1978) for diatoms. The N content was calculated based on a C:N ratios of 6 for *Trichodesmium* (Carpenter et al., 2004), 5 for UCYN-B (Dekaezemacker and Bonnet, 2011; Knapp et al., 2012), and a typical Redfield ratio of 6.6 for

diatoms.

For *Synechococcus* and *Prochlorococcus* we used the C content reported in Buitenhuis et al. (2012) (255 and 36 fg C cell$^{-1}$, respectively), for nano-eukaryotes we used the C content reported in Grégori et al. (2001), and converted into N content according to the C:N Redfield ratio of 6.6 (leading to 3.2, 0.45 and 219 fmol N cell$^{-1}$ for *Synechococcus*, *Prochlorococcus* and nano-eukaryotes). For bacteria, an average N content of 2.1 fg N cell$^{-1}$ (Fukuda et al., 1998) was used.

For the pico-eukaryotes, the cellular N content of $9.2 \pm 2.9$ fmol N cell$^{-1}$ was used as reported in Gregori et al. (2001).

The cell-specific N$_2$ fixation rates and DDN transfer rates (in nmol N L$-1$ 48 h$-1$) to non-diazotrophic phytoplanktonwas calculated for each plankton group analyzed as follows:

$$DD^{15}N = \frac{^{15}Nex}{Nsr} \times N_{con} \times A$$

where $^{15}$Nex (atom %) is the excess $^{15}$N enrichment of the individual cells measured by nanoSIMS after 48 h of incubation

relative to the time zero value, N$_{sr}$ (atom %) is the excess $^{15}$N enrichment of the source pool (N$_2$) in the experimental bottles determined by MIMS, N$_{con}$ is the cellular N content (in nmol N cell$^{-1}$) of each cell and A is the abundance of the specific plankton group (in cell L$^{-1}$). Incertitude was estimated on each variable and the final incertitude was estimated using the propagation error rule.

**2.9 Experimental setup for DDN transfer experiments in zooplankton**

The DDN transfer to zooplankton was measured in four experiments performed at the same stations where the E1, E2 and E3 experiments were performed (hereafter names Zoo-1, Zoo-3 and Zoo-4) plus an additional station located between LDA and LDB (SD9, 20°57'S – 178°39'E). *Trichodesmium* was dominating the diazotroph community in the Zoo-1, Zoo-2, and Zoo-3 experiments, and UCYN-B were dominating the Zoo-4 experiment. The experiments consisted in incubations of freshly collected zooplankton in the presence of the natural planktonic assemblage pre-labelled with $^{15}$N$_2$. In parallel to the

experiments describes above, 6 additional HCl-washed-sample rinsed (three times) 1 L polycarbonate bottles equipped with septum caps were collected by the underway pumping system at 6 m-depth for Zoo-1, Zoo-2 and Zoo-3 and with Niskin bottles at 55 m-depth for Zoo-4. All bottles were amended with 1 mL of $^{15}$N$_2$ (98.9 atom% $^{15}$N, Cambridge isotopes). The bottles were shaken 30 times to facilitate the $^{15}$N$_2$ dissolution and incubated in on-deck incubators for 24 h-36 h as described above.

The incubation was stopped by filtering the bottles on 0.2 µm pore size 47 mm membrane filters and re-suspending the particulate matter in 6 1 L bottles filled with 0.2 µm filtered surface seawater collected at the same station, in such a way that the $^{15}$N enrichment of the food source provided to zooplankton (hereafter referred to as $^{15}$N pre-labelled plankton) stop to



increase by fixing $^{15}N_2$. Meanwhile, zooplankton was collected using repeated net tows (120 mesh size) before dawn. Animals were recovered on a 120 µm sieve and placed into 4.5 L polycarbonate bottles filled with 0.2 µm filtered surface seawater in the dark for at least 6 h in order to allow them to empty their guts. Living animals were visually identified and the individuals belonging to the genus *Clausocalanus*, which largely dominated the zooplankton community at all stations

(Carlotti et al., this issue) were handpicked and 12 animals were dispatched into each of the three 1 L bottles containing the $^{15}N$ pre-labelled plankton before being incubated in on-deck incubators for 24 h as described above. The three other bottles were immediately filtered after the introduction of animals, first through a 120 µm mesh in order to recover the animals and secondly through precombusted (4 h, 450 °C) GF/F filters, which were used to quantify the isotopic signature of the $^{15}N$ pre-labelled plankton at the beginning of the experiment, together with the initial $NH_4^+$ concentrations in the incubation bottles.

After 24h, the triplicate bottles containing the mixture of $^{15}N$ pre-labelled plankton and zooplankton were filtered in the same way. In addition, the filtrate was recovered as described above in order the measure the $NH_4^+$ concentration and the $^{15}N$ enrichment in the dissolved pool by using the two steps diffusion method as described in section 2.3. The recovered animals were placed on GF/F filters, which were analyzed by EA-IRMS as described above in section 2.2.

## 3 Results

### 3.1 Description of the biogeochemical context at the studied stations

Regarding the chlorophyll a and nutrient concentrations, the four studied stations were divided into two main sub-regions (Table 1): i) stations of E1/Zoo-1, Zoo-2 and E2/Zoo3 were located in the oligotrophic MA waters characterized by chlorophyll a concentrations of 0.159-0.377 µg $L^{-1}$ and $NO_3^-$ and $PO_4^-$ concentrations below 50 nmol $L^{-1}$, ii) station E3/Zoo-4 located in the ultra-oligotrophic GY waters presenting lower chlorophyll a concentration than in MA waters (0.053 µg $L^{-1}$),

$NO_3^-$ concentration below 50 nmol $L^{-1}$, and $PO_4^-$ concentration ~110 nmol $L^{-1}$.

*Trichodesmium* dominated the diazotroph community (95 − 100 % of total nifH gene copies detected by qPCR, Stenegren et al., this issue) at 6 m-depth in the MA waters (E1/Zoo-1, Zoo-2 and E2/Zoo3), while UCYN-B dominated (82 % of nifH gene copies) at 55 m-depth in the GY waters (E3/Zoo-4 experiment). MA waters were characterized by a higher abundance of pico-phytoplankton (*Synechococcus* and *Prochlorococcus*) and bacteria abundances (125-200 and 271-424

$10^{11}$ cells $m^{-2}$, respectively) than GY waters (~110 and ~290, $10^{11}$ cells $m^{-2}$, respectively; Bock et al., this issue). In both regions, *Prochlorococcus* dominated pico-phytoplankton biomass (Table 1).

### 3.2 $N_2$ fixation and DDN release in the dissolved pool

Net $N_2$ fixation rates were 20.1 ± 13.4, 49.9 ± 2.4 and 3.2 ± 0.3 nmol N $L^{-1}$ 48 $h^{-1}$ for the E1, E2 and E3 experiments, respectively (Fig. 2). The DDN released to the dissolved pool (the sum of DON and $NH_4^+$) was 14 ± 9 and 8 ± 2 nmol N $L^{-1}$

48 $h^{-1}$ in E1 and E2, and was below quantification limits in E3. Considering gross $N_2$ fixation as the sum of net $N_2$ fixation and DDN release rates (Mulholland et al., 2004), the DDN released in the dissolved pool accounted for 40 ± 27 and 14 ± 4 %



of gross $N_2$ fixation in E1 and E2, respectively (Fig. 4). Most of the $^{15}N$ released was under the form of DON, which accounted for ~93 and ~96 % of the total N release.

### 3.3 Cell-specific $^{15}N$ enrichment and DDN transfer to non diazotrophic plankton

NanoSIMS analyses performed on individual trichomes (E1 and E2) and UCYN-B cells (E3) revealed significant (p<0.05)

$^{15}N$-enrichment after 48 h of incubation (Fig. 1 a, b) compared to T0 samples (0.371 ± 0.005 atom%), indicating active $N_2$ fixation during the experiments:  1.946 ± 0.837 atom% (n=32) in E1, 1.523 ± 0.477 atom% (n=25) in E2 and 4.707 ± 0.210 atom% (n=192) in E3 (Fig. 3). Cell-specific $N_2$ fixation rates of *Trichodesmium* were 252.7 ± 50.5 fmol N cell$^{-1}$ 48 h$^{-1}$ in E1 and 341.5 ± 68.3 fmol N cell$^{-1}$ 48 h$^{-1}$ in E2, and cell-specific rates of UCYN-B were 18.6 ± 3.8 fmol N cell$^{-1}$ 48 h$^{-1}$ in E3.

NanoSIMS analyses performed on non-diazotrophic plankton (diatoms and cell-sorted *Synechococcus* (Fig. 1 d),

*Prochlorococcus*, bacteria, pico- and nano-eukaryotes (Fig. 1 e)) also revealed significant $^{15}N$-enrichment as compared to T0 values (p<0.05) (Fig. 3). The $^{15}N$-enrichment of the non-diazotrophic plankton (all groups pooled together) was not statistically different (p>0.05) between E1 and E2. However, it was significantly lower (p<0.05) in E3 compared to E1 and E2.

Over the 48 h of the experiment, 10 ± 2 % of the total DDN was transferred to non-diazotrophic plankton for E1, 7

± 1 % for E2, and 15 ± 3 % for E3 (Fig.4). For the three experiments, DDN was mainly transferred to *Synechococcus*, *Prochlorococcus* and bacteria in the three experiments and contributed approximately to 98, 92 and 99 % of the transfer in E1, E2 and E3, respectively (Fig. 3). The major part of the transfer took place in pico-phytoplankton (*Synechococcus* and *Prochlorococcus*), accounting for 73 ± 15 %, 68 ± 14 % and 65 ± 13 % of the total transfer into non-diazotrophs in E1, E2 and E3, respectively followed by bacteria (25 ± 5 %, 23 ± 5 % and 34 ± 7 %, respectively (Fig. 4). Lastly, 50 ± 40 %, 10 ± 2

% and 40 ± 27 % of the newly fixed $^{15}N_2$ remained in the pool of diazotrophs (corresponding to the major group of diazotrophs detected at each stations and analysed by nanoSIMS (*Trichodesmium* or UCYN-B), other potential diazotrophs that have not been targeted by qPCR (such as diazotrophic heterotrophic bacteria), and other groups of non-diazotrophic plankton to which $^{15}N_2$ was transferred but that were not analyzed by nanoSIMS due to their very low abundance) for E1, E2 and E3, respectively.

### 3.4 DDN transfer to zooplankton

Before incubation with zooplankton, the isotopic enrichment of the $^{15}N$ pre-labelled plankton averaged 1.035 ± 0.091 atom% in the experiments Zoo-1, Zoo-2 and Zoo-3 (dominated by *Trichodesmium*) and 0.385 ± 0.005 atom% in the experiment Zoo-4 (dominated by UCYN-B). After 24 h of incubation with zooplankton, the $^{15}N$ enrichment of the $^{15}N$ pre-labelled plankton decreased. Meanwhile, the $^{15}N$ enrichment of zooplankton increased as compared to T0 values (0.383 atom% on

average) and reached 0.482, 0.376, 0.513 and 0.368 atom% on average in Zoo-1, Zoo-2, Zoo-3 and Zoo-4, respectively. As the $^{15}N$ enrichment of the initial source food ($^{15}N$ pre-labelled plankton) was different between the four stations/experiments, and in order to compare the results obtained among experiments, we normalized the values as the percentage of initial



amount of $^{15}$N atoms in excess in the $^{15}$N pre-labelled plankton transferred to the different compartments (i.e. conserved in the $^{15}$N pre-labelled plankton pool, transferred to the zooplankton pool and to the $NH_4^+$ pool) at the end of the incubation. In the experiments where *Trichodesmium* dominated the diazotroph community (Zoo-1, Zoo-2, and Zoo-3), $19 \pm 7$ % to $48 \pm 21$ % of the initial DD$^{15}$N remained in the phytoplankton pool, $5 \pm 5$ % to $9 \pm 13$ % was transferred to the zooplankton and

$0.4 \pm 0.3$ to $7 \pm 3$ % was transferred to the $NH_{4+}$ pool (Fig. 5). In the Zoo-4 (where UCYN dominated the diazotroph community, Table 1), a greater proportion of DD$^{15}$N was conserved in the phytoplankton ($76 \pm 34$ %) but a greater transfer to the zooplankton was also observed ($28 \pm 8$ %, Fig. 5). The recovery of the initial DD$^{15}$N was comprised between 29 % and 100 %, suggesting that the remaining fraction was released to the DON pool. Interestingly, the recovery of the DD$^{15}$N was in surplus in the Zoo-4 experiment ($112.5 \pm 8.5$%), suggesting that the DD$^{15}$N transfer in the DON pool is close to zero.

**4 Discussion**

**4.1 DDN release to the dissolved pool**

The quantity and quality of N released by diazotrophs to the dissolved pool during $N_2$ fixation potentially plays a key role in shaping the planktonic and microbial food webs. In this study, *Trichodesmium* released $14 \pm 4$ % to $40 \pm 57$ % of the newly fixed N into the dissolved pool, which is in agreement with values reported in the literature for field studies (Mulholland,

2007; Bonnet et al., 2016a). DON accounted for ~95 % of the DDN released by *Trichodesmium* (Fig. 2), which is in agreement with contributions measured in culture (80 - 90 %; Berthelot et al., 2015) and in the field (Berthelot et al., 2016). The low contribution of $NH_4^+$ to the DDN release does not mean that it was not released, but is likely the results of immediate consumption by surrounding plankton as $NH_4^+$ is known to the preferred N source for marine plankton. On the opposite, part of the DON released by *Trichodesmium* was probably uptaken by heterotrophic and mixotrophic plankton

(Bronk et al., 2007), and part was likely refractory (not easily available for organisms), explaining the observed accumulation in the dissolved pool. If not refractory, the DON would likely have been immediately assimilated as the region where these experiments were performed are strongly limited by N availability (Van Wambeke et al., this issue).

In the E1 experiment, we noticed a large variability of $N_2$ fixation and DDN release rates among the three replicates, which explains the high standard deviations (Fig. 2): two replicates exhibited net $N_2$ fixation rates ~25-30 nmol N L$^{-1}$ 48 h$^{-1}$

and DDN release rates ~7-10 nmol N L$^{-1}$ 48 h$^{-1}$, whereas in the third replicate, the DDN release (~24 nmol N L$^{-1}$ 48 h$^{-1}$) exceeded net $N_2$ fixation rates (~5 nmol N L$^{-1}$ 48 h$^{-1}$). This can be attributed to the decline of *Trichodesmium* in this replicate as we counted much more degraded trichomes in the third replicate. This suggests that decaying *Trichodesmium* release DDN more efficiently than healthy *Trichodesmium*, which has already been observed by Bonnet et al. (2016a). This may also explain why the DDN transfer to non-diazotrophic plankton was slightly higher in E1 ($10 \pm 2$ %) than in E2 ($7 \pm 1$ %),

despite both stations were dominated by *Trichodesmuim*.

The DDN released by UCYN-B (E3), was not quantifiable in our study. However, significant DDN transfer into non-diazotrophic plankton was detected ($15 \pm 3$ % of the total fixed N, Fig. 4), suggesting that the DDN released to the




dissolved pool is likely immediately transferred to surrounding communities. Contrary to E1 and E2, DON did not accumulate in the dissolved pool, suggesting either DON is released by UCYN but is more labile than DON released by *Trichodesmium*, or suggesting that UCYN only release $NH_4^+$ (which is immediately uptaken and thus does not accumulate as in *Trichodesmium* experiments). To our knowledge, this is the first report of DDN release in the field in the presence of a

diazotroph community dominated by UCYN-B. Bonnet et al., (2016b) report low release from UCYN-C in coastal waters of the WTSP ($16 \pm 6$ % of total $N_2$ fixation) compared to *Trichodesmium* ($13 \pm 2$ % to $48 \pm 5$ %; Bonnet et al., 2016b). This seems to indicate that the DDN from UCYN is generally lower than the DDN from *Trichodesmium*. Several hypotheses may explain the differences observed between *Trichodesmium* and UCYN: i) as stated above, the DDN from UCYN may be more available than the DDN from *Trichodesmium*, therefore it does not accumulate in the dissolved pool and is thus not measured

by our techniques, ii) the PCD causing *Trichodesmium* bloom demise can enhance the DDN release (Bar-Zeev et al., 2013), iii) the E3 experiment (dominance of UCYN-B) was performed in the ultra oligotrophic waters of the GY where exogenous factors such as viral lyses (Fuhrman, 1999) and sloppy feeding (O'Neil and Roman, 1992b; Vincent et al., 2007) (which ususally enhance N release) are minimal compared to MA waters where the *Trichodesmium* experiments were performed (Bock et al., this issue). Lastly, the DDN release measured here for UCYN-B is close to the one measured in cultures ($1.0 \pm$

$0.3$ % to $1.3 \pm 0.2$ % , Benavides et al., 2013; Berthelot et al., 2015), where the exogenous factors are reduced, which would plead for hypothesis iii).

The DDN release plays a key role in the transfer of N from diazotrophs to the surrounding non-diazotrophs, only it is not a good indicator of the DDN transfer efficiency as we observed that DDN transfer to non-diazotrophs was higher when the release was low (E3) than when it was high (E1 and E2). This has already been observed in coastal waters of the WTSP

by (Berthelot et al., 2016).

### 4.2 DDN transfer efficiency and pathways in the WTSP

Here we report for the first time data on the transfer of DDN to the planktonic food web under contrasting diazotroph community composition in the open ocean. We reveal that $7 \pm 1$ % to $15 \pm 3$ % of the DDN is transferred to the non-diazotrophic plankton (Fig. 4) at short time scales (48 h), which is in the same order of magnitude than the transfer (~10 %)

reported in coastal waters of the WTSP (Bonnet et al., 2016a; Berthelot et al., 2016). In terms of efficiency, despite UCYN-B fix at lower rates compared to *Trichodesmium*, the DDN originating from UCYN-B is more efficiently transferred to non-diazotrophic plankton ($15 \pm 3$ % of total fixed N in the E3 experiment, Fig. 4) compared to the DDN originating from *Trichodesmium* ($10 \pm 2$ % and $7 \pm 1$ % in the E1 and E2 experiments, respectively). This results is in accordance with the fact that we did not detect any accumulation of $^{15}$N-labelled N forms in the dissolved pool (see section above), suggesting a

higher availability of the DDN released by UCYN-B.

Several studies have proven that a fraction of the DDN release is transferred to surrounding non-diazotrophic plankton, and one of them (Bonnet et al., 2016a) conclude that diatoms are the major beneficiaries of the DDN originating from *Trichodesmium* and develop extensively during/after *Trichodesmium* blooms in the coastal WTSP ocean. Despite



*Trichodesmium* is rarely recovered in sediment traps (Chen et al., 2003; Walsby, 1992), these authors hypothesize a potential tight coupling between *Trichodesmium* blooms and export of organic matter as diatoms are efficient exporters of organic carbon to depth (Nelson et al., 1995). In contrast, in the present study, > 90 % of the DDN was transferred to picoplankton (*Synechococcus*, *Prochlorococcus* and bacteria), whatever the station studied (Fig. 4). The cyanobacteria *Synechococcus* and *Prochlorococcus* were the primary beneficiaries (73 ± 15 %, 68 ± 14 % and 65 ± 13 % of the DDN transfer, Fig. 4), which is consistent with (Bonnet et al., this issue) who observed a positive correlation between $N_2$ fixation rates and the abundance of *Synechococcus* and *Prochlorococcus* in the WTSP. We attributed this difference between the present study and the Bonnet et al., (2016a) study to the phytoplanktonic populations present in ambient water at the time of the experiments. In the Bonnet et al., (2016a) study, diatoms were accounting for ~30 % of the non-diazotrophic phytoplankton biomass at T0, whereas, diatoms were quasi absent (1 % of the non-diazotrophic phytoplankton biomass) in our offshore exp. On the opposite, picoplankton was dominating here with *Prochlorococcus* accounting > 60 % of the picoplankton C biomass (Bock et al., this issue). In the present study, diatom abundances were to low at T0 to show a significant increase in 48 h, even if they benefited from the DDN. However, in E2 the diatom abundances were the highest of the three experiments and (de Verneil et al., 2017) mentioned that the bloom observed at LD B was composed of diatoms and *Trichodesmium*, suggesting that *Trichodesmium* contributed to sustain this bloom.

In the E1 and E2 experiments, where *Trichodesmium* was the dominant diazotroph (Stenegren et al., this issue), the DDN was preferentially transferred to *Synechococcus*, while it was preferentially transferred to *Prochlorococcus* in E3 where UCYN-B was the dominant diazotroph (Stenegren et al., this issue). This suggests a possible coupling between *Synechococcus* and *Trichodesmium* as ever mentioned by Campbell et al. (2005), who report higher *Synechococcus* abundances inside *Trichodesmium* blooms compared to surrounding waters, while it is not the case for other plankton groups. This difference may also be linked with the communities present at the time of the experiments: *Prochlorococcus* accounted for ~65 % of pico-phytoplankton in E1 and E2, while it accounted for ~80 % in GY in E3. While the transfer of DDN to *Prochlorococcus* and *Synechococcus* together was roughly equivalent for E1 and E2 (~70 %, Fig. 4), *Synechococcus* abundances increased by 150 % in E1, and *Prochlorococcus* increased by 12 % during the time course of the experiment, while none of the populations increased in abundance in E2 (Fig. 1 Supp. Info.), which is in agreement with Bock et al. (this issue) who report an increase of the grazing pressure with the decrease of the oligotrophic degree as E1 was performed in more oligotrophic waters than E2.

After *Prochlorococcus* and *Synechococcus*, heterotrophic bacteria were the second beneficiaries of the DDN transfer, especially when the diazotroph community was dominated by UCYN-B (34 ± 7 % of the DDN transfer, Fig. 4). In this experiment, bacteria abundances increased by 70 % on average (Fig. 1 Supp. Info), which is in agreement with Berthelot et al., (2016) and Bonnet et al., (2016a; 2016b) who reported similar bacterial increases in the coastal WTSP. When *Trichodesmium* was the dominant diazotroph, 23-25 % was transferred to bacteria, whose abundance increased by 135 % and 15 % in E1 and E2, consistent with Sheridan et al. (2002) who reported higher bacterial abundances in *Trichodesmium* blooms than in surrounding waters. The significant DDN transfer from *Trichodesmium* to bacteria concurs with



*Trichodesmium* and bacteria association that has been largely highlighted in the last decades (Hmelo et al., 2012; Paerl et al., 1989; Sheridan et al., 2002). That is in accordance with Van Wambeke et al. (this issue) who found that $N_2$ fixation fuels 3-35 % of bacterial production in MA waters. Then, we could not discriminate the DDN transfer to pico- and nano-eukaryotes, but as for diatoms, its transfer represented a low contribution to the overall transfer into non-diazotrophs in this region of the open ocean.

## 4.3 Transfer of DDN to zooplankton

Regarding higher trophic levels, the experiments performed here show that the DDN transfer to the major group of zooplankton present in this ecosystem (the copepod *Clausocalanus*) was less efficient (Fig. 5) when the diazotroph community was dominated by *Trichodesmium* (~5-9 %) than when it was dominated by UCYN-B (~28 %). This result is consistent with a previous study based on analogous $^{15}N_2$ labelling method in coastal waters of the WTSP (Hunt et al., 2016), which also report a higher DDN transfer efficiency in the presence of UCYN.

Regarding the DDN transfer from UCYN-B, although the transfer experiments to phytoplankton and bacteria (E3) and zooplankton (Zoo-4) were not performed in the same incubation bottles, they consistently report lower $^{15}N$-enrichments in all the studied pools as compared to the experiments performed when *Trichodesmium* dominated, but in fine, the DDN transfer efficiency was more important in the presence of UCYN. We observed that a larger fraction of DDN was conserved in the UCYN-B pool than in the *Trichodesmium* pool, and a larger part of the DDN was missing (likely associated to the DON pool) with *Trichodesmium* than with UCYN-B (Fig. 5). These observations are consistent with the transfer experiments E1, E2, and E3 which show that *Trichodesmium* released more DDN in the dissolve pool (DON + $NH_4^+$) than UCYN-B. The DDN released in the $NH_4^+$ pool did not presented significant differences between *Trichodesmium* (Zoo-1, Zoo-2 and Zoo-3) and UCYN-B (Zoo-4), and in the four experiments the DDN contribution was low in the $NH_4^+$ pool, as it was immediately assimilated by surrounding organisms as explained in section 4.1. We suggest that the DDN transfer was higher with UCYN than with *Trichodesmium*, since the UCYN-B can be directly grazed due to their small size (2-3 µm), as mentioned in Hunt et al. (2016) who revealed high UCYN abundance in the copepods guts based on qPCR data, while less *Trichodesmium* were measured. This pleads for a direct transfer of DDN from UCYN-B to zooplankton and indirect transfer from *Trichodesmium* through non-diazotrophs. At the ecosystem level, even if the DDN transfer efficiency (~15 %) to zooplankton from UCYN-B is higher than the one of *Trichodesmium*, the ultimate quantity of DDN transferred to secondary producers is higher when *Trichodesmium* dominate, as cell-specific $N_2$ fixation rates of *Trichodesmium* (~250-340 fmol N cell$^{-1}$ 48 h$^{-1}$) are far higher than those of UCYN-B (~19 fmol N cell$^{-1}$ 48 h$^{-1}$). This result is in agreement with the ones of Carlotti et al., (this issue) base on natutal N isotopic measurements, who revealed that ~50-95 % of the zooplankton originates from $N_2$ fixation in the MA waters and ~10-40 % in the GY waters.

Finally, the DDN transferred to zooplankton, either directly or indirectly, may be released in the dissolved pool as $NH_4^+$, providing additional $NH_4^+$ from DDN in the environment that is likely assimilated by organisms in N-depleted waters. Thus, zooplankton N release appears as another DDN transfer pathway to the microbial communities in the WTSP.





## 5. Conclusion and ecological impact in the WTSP

$N_2$ fixation acts as a natural N fertilizer in the ocean, releasing DDN in the dissolved pool, which are available for surrounding marine organisms. To our knowledge, this study provides the first quantification of DDN transfer to phytoplankton, bacteria and zooplankton communities in open ocean waters. The main interest of this study was to compare

DDN transfer and release under contrasting $N_2$ fixation activity and diversity.

Here, we reveal that *Trichodesmium* released more DDN than UCYN-B, but a significant part of the DDN released by *Trichodesmium* was refractory, while the DDN released by UCYN-B was more bio-available ($NH_4^+$ and labile DON) and likely immediately assimilated by the surrounding plankton communities. The DDN transfer efficiency was more important to non-diazotrophic phytoplankton and bacteria (~15 ± 3 %) and zooplankton (~28 ± 8 %) when UCYN-B dominated the

diazotroph community than when *Trichodesmium* dominated (~8 ± 2 % and 7 ± 6 % of transfer to phytoplankton and bacteria, and zooplankton, respectively). In the open ocean, most of the $N_2$ fixation is performed by *Trichodesmium* (Capone et al., 1997), thus on a global scale most of the DDN transfer can be attributed to *Trichodesmium*, moreover in the MA waters where *Trichodesmium* dominated diazotroph community. The regions where UCYN are the dominant diazotrophs present lower $N_2$ fixation rates than the one of *Trichodesmium*, but it is not negligible and may provide a continuous

background of DDN to surrounding plankton communities. The DDN was preferentially transferred to pico-plankton which is the most abundant plankton community in the WTSP, suggesting that $N_2$ fixation fueled the growth of biomass in the N-depleted environment. That is constant with (Caffin et al., 2017) who revealed that $N_2$ fixation provided more than 90 % of the new N in the photic layer subsequently transformed into bio-available through DDN release, and indicated that $N_2$ fixation contributed to 15-21 % of the PP in the MA waters and ~4 % in the GY waters. On a larger scale view, the

simulation performed by Dutheil et al. (this issue) predicts that diazotrophs support a large part of PP (~15 %) in LNLC regions of the Pacific Ocean, comprising the WTSP.

Overall, this clearly indicates that in the WTSP the $N_2$ fixation plays a key role on the marine biomass production, subsequently on the planktonic food web associated, and finally on the export toward the deep ocean. The DDN can be exported to the deep ocean by different way: i) direct of export of diazotrophs, ii) export of non-diazotrophs which benefit from the DDN transfer, and iii) export of zooplankton which benefit from the DDN transfer. The direct export quantification

in the WTSP, indicates a direct carbon export associated to diazotrophs of ~ 30 % at LD A (E1), 5 % at LD B (E2) and < 0.1 % at LD C (E3) (Caffin et al., 2017). The low [15]N-enrichment of the particulate matter recovered in sediment trap deployed at LD A, LD B and LD C indicates that $N_2$ fixation significantly contributed to particulate export (Knapp et al., this issue), either by direct or indirect export, in the WTSP. Thus, $N_2$ fixation has ineluctably a key role on the biological carbon pump,

as mentioned in Moutin et al. (this issue) who reveal a significant biological "soft tissue" carbon pump in the MA water sustained almost exclusively by $N_2$ fixation, and acting as a net sink for atmospheric $CO_2$.





**Acknowledgements**

This is a contribution of the OUTPACE (Oligotrophy from Ultra-oligoTrophy PACific Experiment) project (https://outpace.mio.univ-amu.fr/) funded by the French research national agency (ANR-14-CE01-0007-01), the LEFE-CyBER program (CNRS-INSU), the GOPS program (IRD) and the CNES (BC T23, ZBC 4500048836). The OUTPACE

5  cruise (http://dx.doi.org/10.17600/15000900) was managed by the MIO (OSU Institut Pytheas, AMU) from Marseille (France) and received funding from European FEDER Fund under project 1166-39417. The authors thank the crew of the R/V L'Atalante for outstanding shipboard operation. G. Rougier and M. Picheral are warmly thanked for their efficient help in CTD rosette management and data processing, as is Catherine Schmechtig for the LEFE CYBER database management. The satellite-derived data of Sea Surface Temperature, chl a concentration and current have been provided by CLS in the

10  framework of the CNES funding; we warmly thank I.Pujol and G.Taburet for their support in providing these data. We acknowledge NOAA, and in particular R.Lumpkin, for providing the SVP drifter. Argo DOI (http://doi.org/10.17882/42182)
All data and metadata are available at the following web address: http://www.obs-vlfr.fr/proof/php/outpace/outpace.php
Aurelia Lozingot is acknowledged for the administrative work.



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



**Table 1: Environmental conditions at stations where experiments were performed. Station position, depth of sampling, concentrations of Chl a, NO$_3^-$, NH4+, DON, PO$_4^-$, DOP, dominant phytoplankton communities and dominant diazotrophs.**

| Experiment | | E1 / Zoo-1 | Zoo-2 | E2 / Zoo-3 | E3 / Zoo-4 | References |
|---|---|---|---|---|---|---|
| Position | Lat. - Lon. | 19°12.8'S 164°41.3'E | 20°57'S 178°39'E | 18°14.4'S 170°51.5'W | 18°25.2'S 165°56.4'W | |
| Depth | m | 6 | 6 | 6 | 55 | |
| Chl a | µg L$^{-1}$ | 0.197 | 0.159 | 0.377 | 0.053 | Dupouy et al., this issue |
| NO$_3^-$ | nmol L$^{-1}$ | 30 | < 10 | 20 | 20 | |
| NH$_4^+$ | nmol L$^{-1}$ | 5 | 2 | 8 | 1 | |
| DON | µmol L$^{-1}$ | 6.20 | 5.20 | 6.00 | 5.15 | |
| PO$_4^-$ | µmol L$^{-1}$ | 10 | 10 | 20 | 110 | |
| DOP | µmol L$^{-1}$ | 0.15 | 0.17 | 0.17 | 0.15 | |
| *Prochlorococcus* | 10$^{11}$ cells m$^{-2}$ integrated on the upper photic layer | 122 ± 31 | | 183 ± 27 | 110 ± 9 | Bock et al., this issue |
| *Synechococcus* | | 3 ± 2 | | 16 ± 9 | 0.5 ± 0.2 | |
| Bacteria | | 271 ± 73 | | 424 ± 108 | 290 ± 32 | |
| Dominant diazotroph | % nifH gene copies | *Trichodesmium* 95 % | *Trichodesmium* 99 % * | *Trichodesmium* 100 % | UCYN-B 82 % | Stenegren et al., this issue |





**Figure captions**

**Figure 1: NanoSIMS images showing the $^{15}$N-enrichment (a,b,d,e) after 48 h of incubation in the presence of $^{15}$N$_2$ for *Trichodesmium* (a), UCYN-B (b), Nano-Eukaryotes (d) and *Synechococcus* (e). NanoSIMS images showing the secondary electrons channel of UCYN (e) and optical camera image of *Prochlorococcus* spotted on the filter before NanoSIMS analyses (f).**

5 **Figure 2: N$_2$ fixation rates (dark grey, nmol N L$^{-1}$ 48 h$^{-1}$), DDN release nmol N L$^{-1}$ 48 h$^{-1}$) as NH$_4^+$ (light grey) and DON (white) for each experiment (E1, E2 and E3). Error bars represent the standard deviation of triplicate incubations.**

**Figure 3: Left panels: box-plot of the 15N-enrichment measured in diazotrophs (*Trichodesmium* for E1 and E3, and UCYN-B for E3). Right panels: $^{15}$N-enrichment measured in non-diazotrophic plankton: *Synechococcus*, *Prochlorococcus*, Bacteria, Diatoms, Pico-Eukaryotes and Nano-Eukaryotes for each experiment. Black dotted line indicates the natural isotopic enrichment.**

10 **Figure 4: DDN fate after 48 h for each experiment. Left pie charts: Orange: DDN remained in diazotrophs (orange), yellow: DDN released to the dissolved pool, Dark blue: DDN transferred to non diazotrophic plankton Right pie charts, from dark blue to light blue: Relative DDN transferred to *Synechococcus*, *Prochlorococcus*, bacteria, diatoms, pico-eukaryotes and nano-eukaryotes in E1 (top), E2 (middle) and E3 (bottom pie chart)..**

**Figure 5: DD$^{15}$N transferred (%) in the NH$_4^+$ pool (white), zooplankton (light grey) and remained in the phytoplankton pool (dark 15 grey) after 24 h of incubation. Error bars represent the standard deviation of triplicate incubations and the propagated analytical errors**



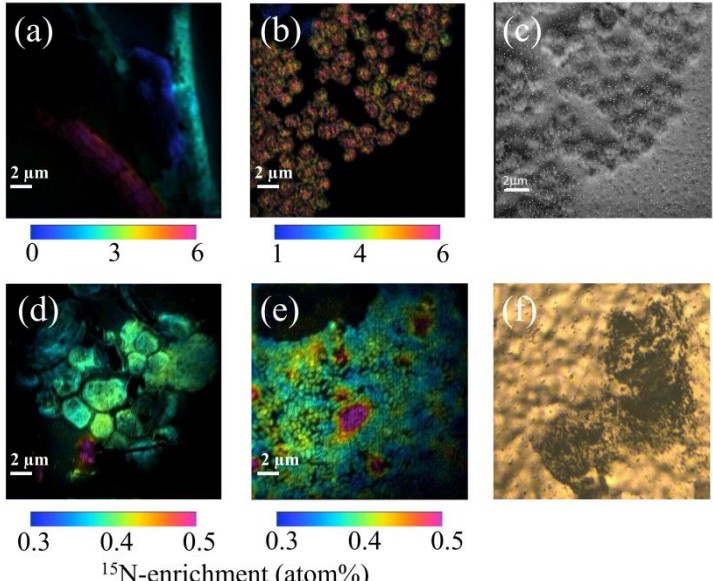

**Figure 1**

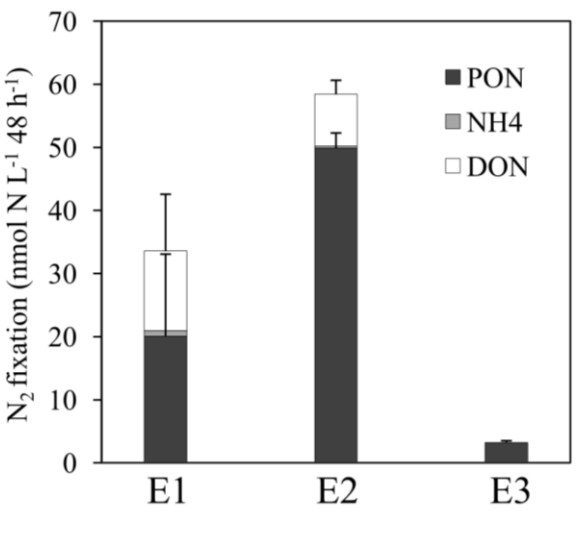

**Figure 2**





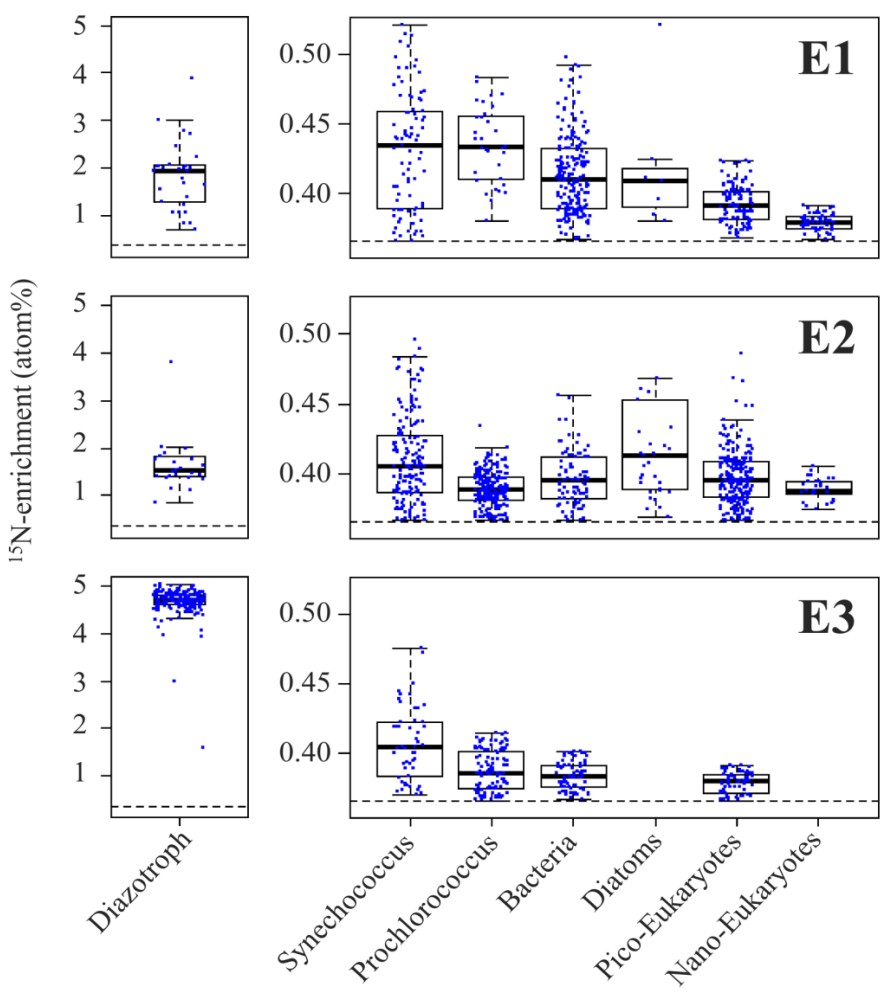

**Figure 3**



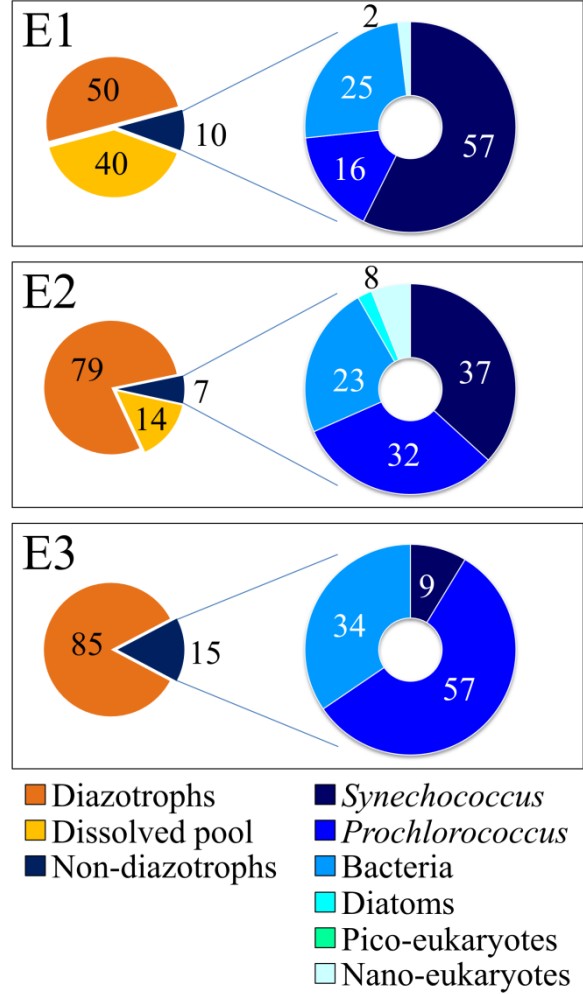

**Figure 4**



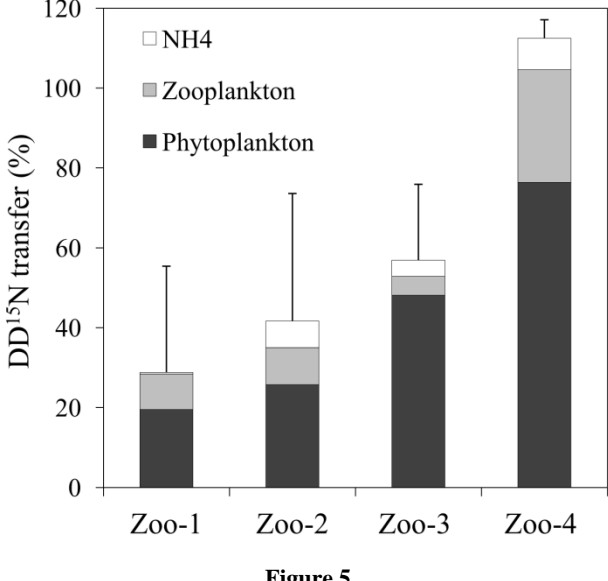

**Figure 5**