# Peer review of "Transfer of diazotroph-derived nitrogen to the planktonic food web across gradients of $N_2$ fixation activity and diversity in the Western Tropical South Pacific"

_Biogeosciences, 2017_

## Referee Comment (RC1) · C. Löscher (Referee) · 6 Feb 2018

The manuscript by Caffin et al. addresses the important question on how much fixed N is transferred to the dissolved versus the particulate planktonic pool. Caffin et al come up with a nanoSIMS based study to not only make this distinction, but to also show that the composition of the diazotrophic community has an impact on the subsequent channeling of N in the Ocean, and they could identify that Trichodesmium promotes a transfer to the dissolved phase, while UCYN-B would promote transfer to non-diazotrophic plankton (mostly picocyanobacteria, followed by heterotrophs). Intriguingly, a higher share of the N pool was transferred to higher trophic levels when Trichodesmium dominated, however, an overall high transfer efficiency was observed in UCYN-B dominated environments. The manuscript is, to my knowledge, one of the first to address the channeling of N through the food web, with that it critically advances the understanding of N2 fixation in the Ocean. I thus highly recommend publication after addressing the following general and specific recommendations.

General comments:

Overall, the manuscript seems to need a bit of streamlining. I see, this is not an easy job to do and I appreciate the thorough introduction and methodological explanations, as well as the detailed description of the results. However, it seems a bit of an overkill given the obvious key results of the two modes of DDN channeling and its subsequent transfer to higher trophic levels. I recommend to reduce the length of the text in order not to dilute your findings.

In the context of the discussion of DDN transferred to zooplankton, either directly or indirectly, I would like to see a link to export production, which may be extremely important in the context of enhanced CO2 uptake through certain ecosystem compositions.

I am a bit worried about two things: first, some share of what you measured may be an artefact due to Trichodesmium's sensitivity to mechanical stress, second, samples were taken using two different methods, i.e. from Niskin bottles and from a pump system, the latter of which is suspected to disrupt cells. Please address those concerns.

Specific comments:

p.1

l. 15: What do you mean with atmospheric- I assume dust input? In a way N2 fixation is atmospheric.

l.16: Which technical limitations- such as tracing the isotope fractionation? That's possible at least to a certain degree

l. 25: this is somewhat difficult to understand as it seems contradictory to the previous sentences. Please clarify that you are referring to the pool that is transferred to plankton

l. 30: Please add an explanation, here, otherwise it seems contradictory to the previous statements

p.2

l.9: Add the study by Duce et al, 2008.

l. 14: I identified some archaea being important in the Pacific, feel free to add the reference (or even not, Löscher et al, 2014 in ISMEj)

l. 31 N2, 2 has to be in subscript

p.4

l.16, l.21: 15N, 15 in upper case

l. 20: Why would Trichodesmium be toxic?

p. 5

l. 15 onwards is largely the exact same text as in ' In depth characterization of diazotroph activity across the Western Tropical South Pacific hot spot of N2 fixation' by Bonnet et al. As there is no point to repeat that, I would recommend to refer to this manuscript instead of having such a strong overlap.

p.8

l.17, l.23, p.9, l.11: please mind the upper and lower cases

p.10

l.28: I would like to see the rates as per day

p.11

l.1 under the form of DON- sounds awkward, please rephrase

l.17 Sentence sounds awkward, please rephrase

l. 19 What bacteria? I assume, non-phototrophic ones. . .please clarify.

l. 29 down to what?

p.11

l.7: I don't quite get this conclusion.

p.12

l.5 + in upper case

l.27 This is actually worrying, thus all of it may be an effect of how Trichodesmium is treated during the experiments

---

## Referee Comment (RC2) · Anonymous Referee #2 · 9 Feb 2018

In this manuscript Caffin et al. examine transfer of diazotroph-derived (DDN) through the foodweb using 15N stable isotope probing, comparing sites dominated by Trichodesmium with a site dominated by UCYN-B as the dominant diazotroph. They find that over 48h in the UCYN-B dominated station, no DDN was detectable in the dissolved pool, whereas a significant fraction was detectable in the Trichodesmium stations. They further characterize DDN to different microbial and zooplankton groups, and find differences between the stations. These results have major ecological implications for our understanding of DDN fate. Overall, I thoroughly enjoyed the manuscript,

and highly recommend it for publication. I do have a few general questions and suggestions regarding the interpretation of the results and the context those results are put in. I recognize that putting these results in context of the other research done on the same cruise is difficult to carve out one piece to focus on, but I think the manuscript could use some focusing.

- Regarding whether Tricho releases recalcitrant N and UCYN-B releases labile N, I'm not sure the data really tells us this. It might mostly be a matter of semantics, and how you define labile and recalcitrant. But for me those terms imply different molecules released by the diazotrophs. From the data I don't think we can rule out that Tricho and UCYN-B release the exact same molecules of N, but because of the difference in both the amount of N released and the composition and metabolic state of the resident community, you see different DDN transfer and efficiency. In fact, I think it's interesting, although maybe expected, that you see higher efficiency in the ultra-oligotrophic location, implying that that maybe that community have higher affinity responses and uptake relative to the resident community in the Tricho stations. Prochlorococcus, for example, is likely to be better at high affinity uptake than Synechococcus because of its smaller surface area to volume ratio and adaptation to oligotrophic environments. Maybe this knowledge could help us predict, by knowing community composition and amount of N fixed, how efficient DDN transfer will be?

- One of the points that the authors emphasize is novel is that this is the first open ocean study. But I am not getting the full context for moving to the open ocean-what do the authors expect will be different, other than diazotroph identity? If this is the focus, it would be nice to include an expectation in the introduction–do they expect the open ocean DDN transfer to be different from the other studies of coastal or mesocosms performed before by this group? Or the same? For example, P.4 line 15-what was expected, different or similar to what found for coastal? Also P.4 lines 25-27. Then, I think these experiments help give us a context to predict DDN transfer through the food web, so I would like some more discussion in that context at the end: i.e. Will we need

to know both diazotroph identity and nutrient conditions to predict DDN transfer? Or other factors? In some ways focusing on "first time in the open ocean" might actually even sell the results a little bit short-is this maybe the first full food web study in this manner as well?

I also have some specific questions and suggestions:

P.8 lines 1-8-Flow sorting before analysis–I would like more information on this method included, when I looked up the referenced Bonnet et al, 2016b, it didn't include flow sorting-is there another paper with these details? If not, more information should be provided in this manuscript in order to verify that you had what was expected on the filter, and the NanoSIMS analysis was on the expected cells. For example, was there any correlated imaging of the filters (i.e. with fluorescence or SEM) to verify and map the cells other than the CCD camera on the NanoSIMS? It would be good to include some more raw data in supplemental with some examples of the NanoSIMS ion and secondary electron images for each group with examples of how ROIs were drawn. Particularly, it seems like the bacteria may have come through in the other sorts, was that a problem and were those identifiable in the NanoSIMS? Prochlorococcus and bacteria for example, might would look similar in the CCD camera?

P.8 line 24-25, a table of ROIs per sample in supp would help, i.e. n for each analysis

p.8 line 32-UCYN-B cell diameters from NS images-interesting and not typical-an example in supp would help, was it correlated with other imaging? (i.e. fluor or SEM?).

P.11 line 5-20 3.3-I couldn't find the information on the T0 values, how many and how analyzed? Everything is relative to the T0 but unclear what the n is.

P.11 Line 15-Sentence "For the three experiments.." -I don't get what this statement means and not sure how it relates to Figure 3

P.11 line 26-Again, like the T0, how was the prelabelled plankton measured? NanoSIMS or IRMS? what is the n?

P.12 lines 3-4-when the error is bigger than the reported number, I worry this becomes meaningless to report-how else can the data be described?

P12-Because averaging to T0, lose some information about total N-fixation. Maybe Zoo4 is only different because lower total enrichment?

P13 line 4-5: but the DDN in the dissolved pool doesn't show release by UCYN-B, the results do imply release because you see DDN transfer but then shouldn't this statement be in the next section?

P15 Line 29-Not clear what that 50-95

P16 line 8 "The DDN transfer efficiency was more important..." not sure what is meant by "more important" more important how?

P.16-last paragraph is a bit confusing and tangential to me. This is just a suggestion, but I would prefer more of a wrap-up on what this data presented means in the context of DDN transfer prediction, e.g. does this help to reconcile the differences between the culture and field studies, or coastal vs. open ocean? What are the implications from the results for predicting transfer through the food web in other areas?

Figure 1: I think in the figure legend "secondary electrons channel of UCYN (e)" should be (c)? Also, does f correlate with anything? Is there a NanoSIMS image of Prochloro-cococus cells?

Figure 4: The left pie charts numbers I think should correspond to P.11 lines 19-20 numbers-but they don't-how much N stays with the diazotrophs? Is it 50, 79 and 85

Technical corrections:

P3 Line 15-16-this sentence is confusing to me, lower than what? In the field?

P9 line 21-22 after "Plus an additional..." add "Zoo-2", if that is what that experiment is, confusing.

---

## Author Response (AR1)

Marseille, 17/05/2018

Dr Douglas G. Capone
Associated Editor – *Biogeosciences*

Dear Associate Editor,

We are very grateful for the opportunity to revise our manuscript entitled *'Transfer of diazotroph-derived nitrogen to the planktonic food web across gradients of N₂ fixation activity and diversity in the Western Tropical South Pacific'*, for publication in *Biogeosciences*. We followed the suggestions from the reviewers, who we thank for the time and effort devoted to the review of the manuscript. Here we send you responses to the reviewers and the revised version of the manuscript in which we have used the 'track changes' mode with different colour for each reviewer to better assess the progress of the manuscript.

Also, we have added Aude Barani as co-author of the manuscript as she performed the cell sorting by flow cytometry.

We thank you for your attention to our manuscript and we hope that you will find this new version ready for publication in *Biogeosciences*.

Sincerely, and on behalf of all co-authors,

Mathieu Caffin

Response to Anonymous Referee #2

We thank Anonymous Referee #2 for the time and effort devoted to the review of the manuscript. Below, we reproduce the reviewer's comments and address their concerns point by point. The reviewer's comments are copied below in regular font with our responses in red. Manuscript changes are shown with additions in bold, deletions in strikethrough.

In this manuscript Caffin et al. examine transfer of diazotroph-derived (DDN) through the foodweb using 15N stable isotope probing, comparing sites dominated by Trichodesmium with a site dominated by UCYN-B as the dominant diazotroph. They find that over 48h in the UCYN-B dominated station, no DDN was detectable in the dissolved pool, whereas a significant fraction was detectable in the Trichodesmium stations. They further characterize DDN to different microbial and zooplankton groups, and find differences between the stations. These results have major ecological implications for our understanding of DDN fate. Overall, I thoroughly enjoyed the manuscript, and highly recommend it for publication. I do have a few general questions and suggestions regarding the interpretation of the results and the context those results are put in. I recognize that putting these results in context of the other research done on the same cruise is difficult to carve out one piece to focus on, but I think the manuscript could use some focusing.

- Regarding whether Tricho releases recalcitrant N and UCYN-B releases labile N, I'm not sure the data really tells us this. It might mostly be a matter of semantics, and how you define labile and recalcitrant. But for me those terms imply different molecules released by the diazotrophs. From the data I don't think we can rule out that Tricho and UCYN-B release the exact same molecules of N, but because of the difference in both the amount of N released and the composition and metabolic state of the resident community, you see different DDN transfer and efficiency. In fact, I think it's interesting, although maybe expected, that you see higher efficiency in the ultra-oligotrophic location, implying that that maybe that community have higher affinity responses and uptake relative to the resident community in the Tricho stations. Prochlorococcus, for example, is likely to be better at high affinity uptake than Synechococcus because of its smaller surface area to volume ratio and adaptation to oligotrophic environments. Maybe this knowledge could help us predict, by knowing community composition and amount of N fixed, how efficient DDN transfer will be?

We agree with the proposition of reviewer #2, that the differences that we observed in DDN release and transfer between the stations is probably the result of contrasted planktonic communities having different affinities due to the trophic state of the station. We have thus taken account to this proposition and discussed this point in the section '4.1 DDN release to the dissolved pool' which is now presented as one of the hypotheses explaining the discrepancy between the two diazotrophs:

[revised manuscript text omitted]

As stated by reviewer #2, this knowledge of the affinity responses of surrounding organisms, by knowing community composition and amount of N fixed, could help us to predict how efficient DDN transfer will be. We agree with this comment and we encourage the scientific community to perform further studies on the point to help understanding and prediction the DDN transfer efficiency.

- One of the points that the authors emphasize is novel is that this is the first open ocean study. But I am not getting the full context for moving to the open ocean-what do the authors expect will be different, other than diazotroph identity? If this is the focus, it would be nice to include an expectation in the introduction–do they expect the open ocean DDN transfer to be different from the other studies of coastal or mesocosms performed before by this group? Or the same? For example, P.4 line 15-what was expected, different or similar to what found for coastal? Also P.4 lines 25-27.

Our group performed similar studies to understand this DDN transfer in coastal water of the WTSP and measured specific transfer rates to the surrounding planktonic communities, but this study provides the first observation of these processes in the open ocean. Previous studies had shown differences of N release between culture and coastal field experiment suggesting a strong influence of the environment on the DDN release. We thus expected to see differences between coastal and open ocean waters in term of release and subsequently in term of transfer. We understand that reviewer #2 would like to see these expectations in the introduction section, thus we have modified the text in the following way: "***The differences of DDN release and transfer rates observed between the different field experiments and the different diazotrophs suggest that these processes strongly depend on the physiological state of diazotrophs and the environment. Yet, To date** the transfer of DDN to different groups of plankton from different diazotroph (Trichodesmium vs. UCYN) in the open ocean, **where most of global marine $N_2$ fixation takes place,** has never been investigated*."

Then, I think these experiments help give us a context to predict DDN transfer through the food web, so I would like some more discussion in that context at the end: i.e. Will we need to know both diazotroph identity and nutrient conditions to predict DDN transfer? Or other factors? In some ways focusing on "first time in the open ocean" might actually even sell the results a little bit short-is this maybe the first full food web study in this manner as well?

We agree with the fact that this study gives us a first estimate of the magnitude of the DDN transfer through the food web in the open ocean. Our long-term goal would be to be able to give solid parametrization for models to predict DDN transfer through the food web. This work has been initiated using a 1-D vertical biogeochemical mechanistic model (Gimenez et al., 2016, https://www.biogeosciences.net/special_issue193.html). However, we think that it might be an overkill to say that we could be able to predict the DDN transfer thanks to our measurements in coupled physical-biogeochemical models. More studies should be performed to have a wider understanding of the processes, in particular the effect of physical processes (not taken into account here), the effect of the physiological state of diazotrophs, the trophic status of the water mass, the plankton community composition, etc…

I also have some specific questions and suggestions:

P.8 lines 1-8-Flow sorting before analysis–I would like more information on this method included, when I looked up the referenced Bonnet et al, 2016b, it didn't include flow sorting-is there another paper with these details? If not, more information should be provided in this manuscript in order to verify that you had what was expected on the filter, and the NanoSIMS analysis was on the expected cells.

The good reference is Bonnet et al., 2016a. This is a mistake that has been corrected in the new version of the manuscript

Information and protocol about flow sorting are available in the supporting information file of Bonnet et al. (2016a) in the section 'Auto- and heterotrophic picoplankton analysis and sorting by flow cytometry' which is available on

https://aslopubs.onlinelibrary.wiley.com/doi/abs/10.1002/lno.10300 .

We understand that it was not clear in the text, thus we have changed the reference 'Bonnet et al. (2016b)' by 'Bonnet et al. (2016a, **Supp. Info.**).

For example, was there any correlated imaging of the filters (i.e. with fluorescence or SEM) to verify and map the cells other than the CCD camera on the NanoSIMS? It would be good to include some more raw data in supplemental with some examples of the NanoSIMS ion and secondary electron images for each group with examples of how ROIs were drawn. Particularly, it seems like the bacteria may have come through in the other sorts, was that a problem and were those identifiable in the NanoSIMS? Prochlorococcus and bacteria for example, might would look similar in the CCD camera?

We agree with the reviewer that *Prochlorococcus* and heterotrophic bacteria look very similar in the CCD camera of the nanoSIMS but also on a SEM. This is why we use cell sorting to discriminate the different groups, using in autofluoresecence for photosynthtetic cells and SYBR green staining for heterotrophic bacteria. In the new version of the manuscript, we added the figure below in the Supp. Info. showing representative cytograms where populations appeared clearly and were well clustered. This argues for a potentially low level of cross contamination in out samples, even though it cannot be excluded.

[Figure]

**Figure 3: Clustering of planktonic communities by flow cytometry on green fluorescence vs. forward scatter cytograms (left) and red fluorescence vs. forward scatter (right): heterotrophic bacteria (red), *Prochlorococcus* (blue), *Synechococcus* (green), and the pico-eukaryotes (pink)**

We agree that it is not clear that how the ROIs were drawn. To clarify this point we have modified the Figure 1 and have added some ROIs on the corresponding images, in the following way.

[Figure]

$^{15}$N-enrichment (atom%)

**Figure 1: NanoSIMS images showing the $^{15}$N-enrichment (a,b,d,e) after 48 h of incubation in the presence of $^{15}$N$_2$ for *Trichodesmium* (a), UCYN-B (b), Nano-Eukaryotes (d) and *Synechococcus* (e). The ROIs are represented in white line.**

**NanoSIMS images showing the secondary electrons channel of UCYN  (c) and optical camera image of *Prochlorococcus* spotted on the filter before NanoSIMS analyses (f).**

In addition we have added complementary nanoSIMS images with ROIs in the Figure 2 of the Supp Info, in the following way:

[Figure]

**Figure 2: NanoSIMS images showing the $^{15}$N-enrichment after 48 h of incubation in the presence of $^{15}$N$_2$ for *Prochlorococcus* (a,b), pico-eukaryotes (c,d), heterotrophic bacteria (e,f), *Synechococcus* (g), and *Trichodesmium* (h). The ROIs are represented in white line.**

P.8 line 24-25, a table of ROIs per sample in supp would help, i.e. n for each analysis

We have added the following table in the Supp. Info. file.

**Table 1: Number of ROIs analyzed for diazotrophs (*Trichodesmium* in E1 and E2, UCYN-B in E3), *Synechococcus*, *Prochlorococcus*, bacteria, diatoms, pico-eukaryotes and nano-eukaryotes, for E1, E2 and E3.**

| Experiment | diazotrophs | *Synechococcus* | *Prochlorococcus* | bacteria | diatoms | pico-euk. | nano-euk. |
|------------|-------------|-----------------|-------------------|----------|---------|-----------|-----------|
| E1 | 32 | 87 | 32 | 200 | 8 | 111 | 60 |
| E2 | 25 | 156 | 213 | 85 | 33 | 200 | 29 |
| E3 | 192 | 50 | 115 | 70 | 0 | 70 | 0 |

p.8 line 32-UCYN-B cell diameters from NS images-interesting and not typical-an example in supp would help, was it correlated with other imaging? (i.e. fluor or SEM?).

The UCYN-B cell diameters from nanoSIMS images were 2.3 ± 0.3 µm. Also, UCYN-B cell diameters, measured using a Zeiss Axio Observer epifluorescence microscope were 2.7 ± 0.3 µm. This has been specified in the '2.7 Cell-specific N content and DDN transfer calculations' section in the following way: *"For UCYN-B, cells diameters were directly measured on the nanoSIMS images **and further confirmed on microscopic images**."*

P.11 line 5-20 3.3-I couldn't find the information on the T0 values, how many and how analyzed? Everything is relative to the T0 but unclear what the n is.

The T0 values were measured on 5 cells and analyzed on the nanoSIMS, with the same protocol as for all the measurements. T0 here are within the range of T0 values reported by Bonnet et al. (2016a; 2016b) et Berthelot et al. (2016). As the sentence is unclear, we have changed the text in the following way: *"...compared to T0 samples (0.371 ± 0.005 atom%, **n=5**), ..."*

P.11 Line 15-Sentence "For the three experiments.." -I don't get what this statement means and not sure how it relates to Figure 3

This statement relates to Figure 4 and not Figure 3. We apologize for this mistake which has been corrected in the new version of the manuscript.

P.11 line 26-Again, like the T0, how was the prelabelled plankton measured? NanoSIMS or IRMS? what is the n?

The T0 of the prelabelled plankton were measured by EA-IRMS for the zooplankton experiments. We performed triplicates for each experiment. We have added a new sentence in the Method section to clarify: *"...stop to increase by fixing $^{15}N_2$. **The initial $^{15}N$ enrichment of the $^{15}N$ pre-labelled plankton was analyzed in triplicates by EA-IRMS.** Meanwhile, zooplankton was collected...".* In addition, we have added the number of measurements corresponding to the T0 in the Results section in the following way: *"Before incubation with zooplankton, the isotopic enrichment of the $^{15}N$ pre-labelled plankton averaged 1.035 ± 0.091 atom% (**n=9**) in the experiments Zoo-1, Zoo-2 and Zoo-3 (dominated by Trichodesmium) and 0.385 ± 0.005 atom% (**n=3**) in the experiment Zoo-4 (dominated by UCYN-B)."*

P.12 lines 3-4-when the error is bigger than the reported number, I worry this becomes meaningless to report-how else can the data be described?

We acknowledge that the reported error is high, potentially due to the *Trichodesmium* decay in one of the replicate bottle, as mentioned in the discussion section. However, we are confident that the results are still meaningful in regard of the transfer efficiency between diazotroph and non-diazotroph plankton.

P12-Because averaging to T0, lose some information about total N-fixation. Maybe Zoo4 is only different because lower total enrichment?

We averaged T0 of Zoo-1, Zoo-2 and Zoo-3 because there were not significantly different. T0 of Zoo-4 was not averaged with the others experiments because the $^{15}N$ enrichment was lower as stated by Reviewer #2. As it is unclear, we have modified the text in the following way: *"Before incubation with zooplankton, the isotopic enrichment of the $^{15}N$ pre-labelled plankton **was not significantly different in the experiments Zoo-1, Zoo-2 and Zoo-3 (dominated by Trichodesmium)** averag ed 1.035 ± 0.091 atom% (**n=9**).  **The isotopic enrichment was lower in the experiment Zoo-4 (dominated by UCYN-B)** averaging 0.385 ± 0.005 atom% (**n=3**).  After 24 h of incubation with zooplankton, the $^{15}N$ enrichment of the $^{15}N$ pre-labelled plankton **decreased down to 0.431 ± 0.014 atom% on average in Zoo-1, Zoo-2 and Zoo-3, and down to 0.372 ± 0.010 atom% in Zoo-4**."*

P13 line 4-5: but the DDN in the dissolved pool doesn't show release by UCYN-B, the results do imply release because you see DDN transfer but then shouldn't this statement be in the next section?

The DDN measured in the dissolved pool can only come from UCYN-B, and thus show that DDN was released by UCYN-B. However, Reviewer #2 is true when mentioning that seeing DDN transfer implies that DDN was previously released. Thus, we agree that this sentence is suitable in both sections but we think that it makes more sense in this section.

P15 Line 29-Not clear what that 50-95

Here, we mean that diazotrophs contributed from 50 to 95 % to zooplankton biomass in the MA waters. As it is unclear, we have modified the text in the following way: "*This result is in agreement with  **Carlotti et al. (this issue) results based on** $^{15}N$ **isotopic data showing** that ~50-95 % **and ~10-40 % of the zooplankton N content**  originates from $N_2$ fixation in  MA  and ~ GY waters**, respectively.**"*

P16 line 8 "The DDN transfer efficiency was more important..." not sure what is meant by "more important" more important how?

By 'important' we mean 'higher'. The sentence has been clarified in the following way "*The DDN transfer efficiency*  **to non-diazotrophic plankton was higher**  ~~to non-diazotrophic phytoplankton and bacteria (~15 ± 3 %) and zooplankton (~28 ± 8 %)(~8 ± 2 % and 7 ± 6 % of transfer to phytoplankton and bacteria, and zooplankton, respectively).~~"

P.16-last paragraph is a bit confusing and tangential to me. This is just a suggestion, but I would prefer more of a wrap-up on what this data presented means in the context of DDN transfer prediction, e.g. does this help to reconcile the differences between the culture and field studies, or coastal vs. open ocean? What are the implications from the results for predicting transfer through the food web in other areas?

To clarify the conclusion section, we have modified the text in the following way:

"*5. Conclusion and ecological impact* **of** $N_2$ **fixation** *in the WTSP*

$N_2$ *fixation acts as a natural N fertilizer in the ocean, releasing DDN in the dissolved pool, which is available for surrounding marine organisms. To our knowledge, this study provides the first quantification of DDN transfer to phytoplankton, bacteria and zooplankton communities in open ocean waters. The main interest of this study was to compare DDN transfer and release under contrasting $N_2$ fixation activity and diversity.*

*Here, we reveal that Trichodesmium released more DDN than UCYN-B, but a significant part of the DDN released by Trichodesmium* **accumulated in the dissolved pool** , *while the DDN released by UCYN-B was*  *immediately assimilated by the surrounding plankton communities. The DDN transfer efficiency*  **to non-diazotrophic plankton was higher** ~~more important to non-diazotrophic phytoplankton and bacteria (~15 ± 3 %) and zooplankton (~28 ± 8 %)(~8 ± 2 % and 7 ± 6 % of transfer to phytoplankton and bacteria, and zooplankton, respectively).of the, moreover in the MA waters where Trichodesmium dominated diazotroph community~~. *The regions where UCYN are the dominant diazotrophs* **generally** *present lower*

*N₂ fixation rates than the ones **where**  Trichodesmium **dominates**, but **UCYN provide a continuous source of DDN to surrounding plankton communities** . The DDN was  transferred to pico-plankton, which **dominated**  the WTSP, suggesting that N₂ fixation fueled the growth of biomass in the N-depleted environment. This is consistent with Caffin et al., (2018), who revealed that N₂ fixation provides >  90 % of the new N **to** the photic layer **of the WTSP** ~~subsequently transformed into bio-available through DDN release, and indicated that N₂ fixation contributed to 15-21 % of the PP in the MA waters and ~4 % in the GY waters~~. On a larger scale view, the simulation performed by Dutheil et al. (this issue) predicts that diazotrophs support a large part of PP (~15 %) in LNLC regions of the Pacific Ocean, comprising the WTSP.*

*Overall, this **study**  indicates that  N₂ fixation plays a key role on the marine biomass production, **the structure of**  the planktonic food web , and finally on the export **of organic matter** towards the deep ocean. The DDN can be exported to the deep ocean by different **path**ways: i) **the** direct of export of diazotrophs, ii) **the** export of non-diazotrophs which benefit**ed** from the DDN , and iii) **the** export of zooplankton which benefit**ed** from the DDN . The direct export **of diazotrophs accounted for**  ~ 30 % **of total C export** at LD A (E1), 5 % at LD B (E2) and < 0.1 % at LD C (E3) (Caffin et al., 2018). **Using a δ¹⁵N budget, Knapp et al., (This issue) found that 50-80 % of exported material was sustained by N₂ fixation (this includes both direct and indirect export of DDN).**  Thus, N₂ fixation has ineluctably a key role on the biological carbon pump, as mentioned in Moutin et al. (this issue) who reveal a significant biological "soft tissue" carbon pump in the MA waters **almost exclusively** sustained  by N₂ fixation, and acting as a net sink  **of** atmospheric CO₂ **in the WTSP**."*

Figure 1: I think in the figure legend "secondary electrons channel of UCYN (e)" should be (c)? Also, does f correlate with anything? Is there a NanoSIMS image of Prochlorococcus cells?

In the figure legend "secondary electrons channel of UCYN (e)" has been corrected by "secondary electrons channel of UCYN (c)". The image (f) does not correlate with any of the other images. NanoSIMS image of *Prochlorococcus* cells were added in the Supp Info as explained above.

Figure 4: The left pie charts numbers I think should correspond to P.11 lines 19-20 numbers-but they don't-how much N stays with the diazotrophs? Is it 50, 79 and 85

Reviewer #2 is right pie charts numbers did not corresponded to P.11 lines 19-20 numbers. This is a mistake that has been corrected in the new version of the manuscript. The right numbers are 50 ± 40 %, 79 ± 4 % and 85 ± 9 %.

**Technical corrections:**

P3 Line 15-16-this sentence is confusing to me, lower than what? In the field?

We agree that this sentence is confusing. The DDN release measured in culture studies is much lower than in field studies. We have clarified this sentence in the following way: "***The DDN release is generally much lower in (<5%) in monospecific cultures (Berthelot et al. 2015, Benavides et al. 2013) than in field experiment***  *suggesting that external factors such as sloppy feeding and viral lysis have a strong influence on the DDN release by diazotrophs in field.*"

P9 line 21-22 after "Plus an additional..." add "Zoo-2", if that is what that experiment is, confusing.

To avoid confusion we have had "Zoo-2" in the sentence, as recommended by Reviewer #2, in the following way: "*...plus an additional station (**Zoo-2**) located between LDA and LDB...*"

We thank Carolin Löscher for the time and effort devoted to the review of the manuscript. Below, we address her concerns point by point. The reviewer's comments are copied below in regular font with our responses in blue. Manuscript changes are shown with additions in bold, deletions in strikethrough.

The manuscript by Caffin et al. addresses the important question on how much fixed N is transferred to the dissolved versus the particulate planktonic pool. Caffin et al come up with a nanoSIMS based study to not only make this distinction, but to also show that the composition of the diazotrophic community has an impact on the subsequent channeling of N in the Ocean, and they could identify that Trichodesmium promotes a transfer to the dissolved phase, while UCYN-B would promote transfer to non-diazotrophic plankton (mostly picocyanobacteria, followed by heterotrophs). Intriguingly, a higher share of the N pool was transferred to higher trophic levels when Trichodesmium dominated, however, an overall high transfer efficiency was observed in UCYN-B dominated environments. The manuscript is, to my knowledge, one of the first to address the channeling of N through the food web, with that it critically advances the understanding of N2 fixation in the Ocean. I thus highly recommend publication after addressing the following general and specific recommendations.

**General comments:**

 Overall, the manuscript seems to need a bit of streamlining. I see, this is not an easy job to do and I appreciate the thorough introduction and methodological explanations, as well as the detailed description of the results. However, it seems a bit of an overkill given the obvious key results of the two modes of DDN channeling and its subsequent transfer to higher trophic levels. I recommend to reduce the length of the text in order not to dilute your findings.

We understand that the length of the text can dilute the findings and have thus followed the recommendation of C. Löscher to streamline our manuscript.

Concerning the introduction, as far as we are aware, there are few studies that address the release and fate of DDN in the ocean and we believe it is important to give a detailed state of the art on this topic to introduce the main goals of the present study.

Regarding the Methods section, we reduced the text as much as possible by referring to articles already published (mainly in the same special issue). We changed the text as follows:.

"

[revised manuscript text omitted]

We agree with this comment, and thus we have added this paragraph at the end of the section *"Zooplankton can contribute to organic matter export by production of sinking fecal pellets, active transport to depth and carcasses export. These processes are increasingly recognized as important vectors of organic matter export, and the magnitude of their contributions to organic matter export are highly dependent on regionally variable plankton community structure (Steinberg and Landry, 2017). In the WTSP, where $N_2$ fixation sustains most of the new primary production (Caffin et al., 2018) and an important fraction of the DDN is transferred to zooplankton, it might play a key role on the export production and hence the $CO_2$ sink which is the WTSP."*

I am a bit worried about two things: first, some share of what you measured may be an artefact due to Trichodesmium's sensitivity to mechanical stress, second, samples were taken using two different methods, i.e. from Niskin bottles and from a pump system, the latter of which is suspected to disrupt cells. Please address those concerns.

The underway surface pump is large volume pumping system with large tubing and we paid careful attention to set the final spigot to a gentle flow rate. We previously checked through microscopic observations that fragile cells like diatoms or *Trichodesmium* colonies were not destroyed by this sampling strategy. However, we cannot explain why one of the replicate of the E1 experiment was so different from the others and cannot exclude a potential bias during the sampling with the pump, although from our experience such discrepancies between replicates have already been observed after sampling with Niskin bottles in *Trichodesmium* blooms. We have acknowledged that in the discussion section (see below). Another

explanation could be related to the spatial distribution of *Trichodesmium*, which is very patchy in the ocean during blooms. We may have sampled different populations in different physiological states in different bottles, which may explain the discrepancy between replicates in the E1 experiment.

We have addressed these concerns in the discussion sections, subsection '4.1 DDN release to the dissolved pool' and modified the text in the following way:

"        ***Conversely to E1 and E2,*** *the DDN released by UCYN-B (E3), was not quantifiable in our study. However, significant DDN transfer into non-diazotrophic plankton was detected (15 ± 3 % of the total fixed N, Fig. 4), suggesting that the DDN released to the dissolved pool is likely immediately transferred to surrounding communities.*  *To our knowledge, this is the first report of DDN release in the field in the presence of a diazotroph community dominated by UCYN-B. Bonnet et al., (2016b) report low release from UCYN-C in coastal waters of the WTSP (16 ± 6 % of total $N_2$ fixation) compared to Trichodesmium (13 ± 2 % to 48 ± 5 %; Bonnet et al., 2016b). This seems to indicate that the DDN from UCYN is generally lower than the DDN from Trichodesmium. Several hypotheses may explain the differences observed between Trichodesmium and UCYN.*  *DDN* **compounds released by**  *UCYN may be more* **bio-***available than the DDN*  **released by** *Trichodesmium,* **limiting its accumulation.**  **The lack of accumulation in E3 could also be due to the more severe N limitation of planktonic communities in the ultra-oligotrophic waters as compared to MA waters (Van Wambeke, this issue), and to the nature of the resident community. Prochlorococcus was dominating the planktonic community at LD C (E3) and is known to have a high affinity to to its small surface to volume ratio (Partenski et al., 1999).**  *PCD causing Trichodesmium bloom demise can* **also be involved in the relatively high**  *DDN* **release and accumulation during Trichodesmium dominated experiments** *(Bar-Zeev et al., 2013).*  **Exogenous factors, such as viral lyses (Fuhrman, 1999) and sloppy feeding (O'Neil and Roman, 1992b; Vincent et al., 2007) are also suspected to enhance the DDN release. These factors were found to excert a higher pressure in the**  *MA waters where*  *Trichodesmium* **dominated compared to ultra-oligotrophic waters**  *(Bock et al., this issue)***, where UCYN-B dominated. Finally, part of the discrepancy might be due to a methodological artefact: different sampling procedures between E1 and E2 (pump) and E3 (Niskin bottles) as the pump is suspected to induce mechanical stress to the cells which may have potentially affected the DDN release.**

        *The DDN release plays a key role ...*"

**Specific comments:**

p.1

l. 15: What do you mean with atmospheric- I assume dust input? In a way N2 fixation is atmospheric.

> In this sentence 'atmospheric' refers to atmospheric deposition. To avoid any confusion we have clarified the sentence in the following way: *"Biological dinitrogen ($N_2$) fixation provides the major source of new nitrogen (N) to the open ocean, contributing more than atmospheric **deposition** and riverine inputs to the N supply."*

l.16: Which technical limitations- such as tracing the isotope fractionation? That's possible at least to a certain degree

> Here, by 'technical limitation' we meant isotope tracking in the different planktonic groups. This technical limitation has been unlocked by the use of nanoSIMS method coupled to cell identification (in situ hybridization or flow cytometry). Nevertheless, the fate of DDN in the planktonic food web is still poorly understood and motivated this study. In order to clarify and keep the abstract concise, we have removed "due to technical limitations".

l. 25: this is somewhat difficult to understand as it seems contradictory to the previous sentences. Please clarify that you are referring to the pool that is transferred to plankton

> We acknowledge that is unclear and seems contradictory to the previous sentences, thus we have removed the sentence.

l. 30: Please add an explanation, here, otherwise it seems contradictory to the previous statements

> We understand that this sentence seems contradictory to the previous statements, in fact we made a mistake in the sentence as we wrote 'more' instead of 'less'. We apologize for this mistake and thus we have corrected the sentence in the following way: *" Regarding higher trophic level, the DDN transfer to the dominant zooplankton species was  **less** efficient when the diazotroph community was dominated by Trichodesmium (~5-9 % of the DDN transfer) than when it was dominated by UCYN-B (~28 ± 13 % of the DDN transfer)."*

> In addition, to be clearer, we have modified the previous sentence in the following way:

p.2

l.9: Add the study by Duce et al, 2008.

> We have added this reference in the new version of the manuscript. In addition, we have specified that 'atmospheric input' in this sentence refers to 'atmospheric deposition' as mentioned in a comment above. Thus, the sentence has been modified in the following way: *"At the global scale, $N_2$ fixation is the major source of new N to the ocean, before atmospheric **deposition** and riverine inputs (100-150 Tg N $yr^{-1}$, **Duce et al., 2008;** Gruber, 2008)."*

l. 14: I identified some archaea being important in the Pacific, feel free to add the reference (or even not, Löscher et al, 2014 in ISMEj)

> We have mentioned the archaea in the new version of the manuscript and added the reference Löscher et al (2014) in the following way: *"$N_2$ fixation is performed by prokaryotic organisms termed diazotrophs, which include the non-heterocystous filamentous cyanobacterium*

*Trichodesmium [...], unicellular cyanobacteria termed UCYN [...],  diverse non-cyanobacterial bacteria [...], **and archaea (Löscher et al., 2014)**."*

l. 31 N2, 2 has to be in subscript

This has been corrected in the new version.

p.4

l.16, l.21: 15N, 15 in upper case

This has been corrected in the new version.

l. 20: Why would Trichodesmium be toxic?

The studies that we referred to in the manuscript mentioned intracellular toxins and toxic compounds into these cyanobacteria that can affect zooplankton.

p. 5

l. 15 onwards is largely the exact same text as in 'In depth characterization of diazotroph activity across the Western Tropical South Pacific hot spot of N2 fixation' by Bonnet et al. As there is no point to repeat that, I would recommend to refer to this manuscript instead of having such a strong overlap.

We agree with this comment and the first general comment above that our manuscript needed a bit of streamlining. Thus we have reduced the length of the Methods section and, here in this 'Net $N_2$ fixation rates' section, we have referred to Bonnet et al. (this issue).

p.8

l.17, l.23, p.9, l.11: please mind the upper and lower cases

This has been corrected in the new version.

p.10

l.28: I would like to see the rates as per day

Our experiments were performed on a 48 h time scale and we performed a N budget to determine where does the fixed N goes after 48 h. This explains why we present $N_2$ fixation rates over 48 h. At all stations, we also measured them after 24 h, and could present them as well, but we should probably discuss why rates at 48 h are not the double of those after 24 h (mortality or growth of diazotrophes, equilibration of the $^{15}N_2$ bubble, etc), which is not the scope of this study. Therefore, to be consistent with the other measurements (nanoSIMS data, release data, etc...) that were presented for 48 h of incubation and keep focus on the main scientific question of this study, we believe that it is more appropriate to present  $N_2$ fixation rates per 48 h.

p.11

l.1 under the form of DON- sounds awkward, please rephrase

We rephrased the sentence as *"**DON accounted for the major part of the $^{15}N$ released and accounted for ~93 and ~96 % of the total N release in E1 and E2, respectively."***

l.17 Sentence sounds awkward, please rephrase

We agree with this comment, thus we have splitted the sentence for more clarity in the following way: *" DDN was mainly transferred to  **(Fig. 4)**.  pico-**cyanobacteria**  (Synechococcus and Prochlorococcus), accounting for 73 ± 15 %, 68 ± 14 % and 65 ± 13 % of the total transfer into non-diazotrophs in E1, E2 and E3, respectively **(Fig. 4)**.  **The transfer into heterotrophic** bacteria **accounted for** 25 ± 5 %, 23 ± 5 % and 34 ± 7 % **of the total transfer, in E1, E2 and E3,** respectively ."*

l. 19 What bacteria? I assume, non-phototrophic ones. . .please clarify.

This is heterotrophic bacteria. This has been clarified in this sentence (see response above).

l. 29 down to what?

We have specified the final $^{15}$N enrichment in the new version. Thus we have modified the sentence in the following way: *"After 24 h of incubation with zooplankton, the $^{15}$N enrichment of the $^{15}$N pre-labelled plankton **decreased down to 0.431 ± 0.014 atom% on average in Zoo-1, Zoo-2 and Zoo-3, and down to 0.372 ± 0.010 atom% in Zoo-4**."*

p.11

l.7: I don't quite get this conclusion.

Here we provide the averaged cell-specific $N_2$ fixation rates for *Trichodesmium* in E1 and E2, and UCYN-B in E3. These rates were calculated as:

Cell-specific $N_2$ fixation rates = $(^{15}N_{ex} \times N_{cont}) / N_{sr}$

Where $^{15}N_{ex}$ is the cell-specific $^{15}$N enrichment, $N_{cont}$ is the cell-specific N content and $N_{sr}$ is the $^{15}$N enrichment of the source pool ($N_2$) in the bottles.

p.12

l.5 + in upper case

This has been corrected in the new version.

l.27 This is actually worrying, thus all of it may be an effect of how Trichodesmium is treated during the experiments

We are aware that *Trichodesmium* can be affected by the conditions of the experiments which are discussed in the manuscript. The discrepancy can results from PCD which appears to be highly stochastic and is hard to foresee.

[revised manuscript text omitted]
 *Prochlorococcus* (a,b), pico-eukaryotes (c,d), heterotrophic bacteria (e,f), *Synechococcus* (g), and *Trichodesmium* (h). The ROIs are represented in white line.**

[Figure]

**Figure 3: Clustering of planktonic communities by flow cytometry on green fluorescence vs. forward scatter cytograms: heterotrophic bacteria (red),** *Prochlorococcus* **(blue),** *Synechococcus* **(green), and the pico-eukaryotes (pink)**

**Table 1: Number of ROIs analyzed for diazotrophs (*Trichodesmium* in E1 and E2, UCYN-B in E3), *Synechococcus*, *Prochlorococcus*, bacteria, diatoms, pico-eukaryotes and nano-eukaryotes, for E1, E2 and E3.**

| Experiment | diazotrophs | *Synechococcus* | *Prochlorococcus* | bacteria | diatoms | pico-euk. | nano-euk. |
|:---:|:---:|:---:|:---:|:---:|:---:|:---:|:---:|
| E1 | 30 | 87 | 32 | 200 | 8 | 111 | 60 |
| E2 | 25 | 156 | 213 | 85 | 33 | 200 | 29 |
| E3 | 192 | 50 | 115 | 70 | 0 | 70 | 0 |